# Lumbo-pelvic proprioception in sitting is impaired in subgroups of low back pain–But the clinical utility of the differences is unclear. A systematic review and meta-analysis

Vasileios Korakakis[1,2,3,☯,¤a]*, Kieran O'Sullivan[4,5,☯,¤b], Argyro Kotsifaki[1,☯], Yiannis Sotiralis[2‡], Giannis Giakas[3‡]

**1** Aspetar, Orthopaedic and Sports Medicine Hospital, Doha, Qatar, **2** Faculty of Physical Education and Sport Science, University of Thessaly, Trikala, Greece, **3** Hellenic Orthopaedic Manipulative Therapy Diploma (HOMTD), Athens, Greece, **4** School of Allied Health, University of Limerick, Limerick, Ireland, **5** Ageing Research Centre, University of Limerick, Limerick, Ireland

☯ These authors contributed equally to this work.
¤a Current address: Aspetar, Orthopaedic and Sports Medicine Hospital, Doha, Qatar
¤b Current address: School of Allied Health, University of Limerick, Limerick, Ireland
‡ These authors also contributed equally to this work.
* Vasileios.Korakakis@aspetar.com

**Data Availability Statement:** Data are available as Supporting Information files.

## Abstract

### Background

Altered spinal postures and altered motor control observed among people with non-specific low back pain have been associated with abnormal processing of sensory inputs. Evidence indicates that patients with non-specific low back pain have impaired lumbo-pelvic proprioceptive acuity compared to asymptomatic individuals.

### Objective

To systematically review seated lumbo-pelvic proprioception among people with non-specific low back pain.

### Methods

Five electronic databases were searched to identify studies comparing lumbo-pelvic proprioception using active repositioning accuracy in sitting posture in individuals with and without non-specific low back pain. Study quality was assessed by using a modified Downs and Black's checklist. Risk of bias was assessed using an adapted tool for cross-sectional design and case–control studies. We performed meta-analysis using a random effects model. Meta-analyses included subgroup analyses according to disability level, directional subgrouping pattern, and availability of vision during testing. We rated the quality of evidence using the GRADE approach.

**Funding:** Open Access funding provided by the Qatar National Library.

**Competing interests:** The authors have declared that no competing interests exist.

## Results

16 studies met the eligibility criteria. Pooled meta-analyses were possible for absolute error, variable error, and constant error, measured in sagittal and transverse planes. There is very low and low certainty evidence of greater absolute and variable repositioning error in seated tasks among non-specific low back pain patients overall compared to asymptomatic individuals (sagittal plane). Subgroup analyses indicate moderate certainty evidence of greater absolute and variable error in seated tasks among directional subgroups of adults with non-specific low back pain, along with weaker evidence (low-very low certainty) of greater constant error.

## Discussion

Lumbo-pelvic proprioception is impaired among people with non-specific low back pain. However, the low certainty of evidence, the small magnitude of error observed and the calculated "noise" of proprioception measures, suggest that any observed differences in lumbo-pelvic proprioception may be of limited clinical utility.

## PROSPERO-ID

CRD42018107671

## Introduction

Low back pain is a highly prevalent disabling musculoskeletal disorder and represents a significant burden for the society [1, 2]. Approximately 90% of low back pain is non-specific low back pain (NSLBP) where the pain can be directly attributed to pathoanatomical cause [3]. Mounting evidence suggests that NSLBP is not a homogenous group, but rather denotes a variety of clinical presentations which may differ across several variables such as for example: movement behaviors, pain-provoking or pain-relieving movements and postures, pain distribution patterns, alterations in motor control, and psychological aspects [4–6]. A multidimensional classification system has been proposed by O'Sullivan [5] which classifies patients based on pain symptoms and movement behaviors. To illustrate, the system classifies patients into a) the "flexion pattern" (FP) when provocative movements and postures involve spinal flexion, b) the "active and passive extension pattern" (AEP, PEP) when provocative movements and postures involve extension, and c) the "multidirectional pattern" when all movement directions provoke symptoms [5]. Altered spinal postures and altered motor control have been associated with abnormal processing of sensory inputs, such as proprioception, in patients suffering from NSLBP that vary between the subgroups [6–10]. Evidence from recent systematic reviews [11–13] indicate that patients with NSLBP have significantly impaired lumbo-pelvic proprioceptive acuity compared to asymptomatic individuals. Also, consistent findings have been reported [11] of an increased absolute repositioning error (RE) within a NLBP subgroup (flexion pattern) [9, 14–16]. It has been suggested that these proprioceptive deficits might be associated with the underlying characteristics and mechanisms of NSLBP development [17, 18]. However, the precise association between NSLBP and proprioceptive deficits is unclear, as spinal proprioceptive impairments are not correlated with pain and disability [19, 20].

Lumbo-pelvic accuracy and precision has been found to be considerably affected by test position, and REs appear significantly larger in sitting than in standing [21]. Moreover, no

correlation has been demonstrated between tests for kinaesthesia and joint position sense, or
between different position sense tests [22]. Previous systematic reviews with meta-analyses of
cross-sectional studies, demonstrated that patients with NSLBP present greater variability and
error in spinal proprioceptive acuity as compared to asymptomatic individuals [11, 12]. How-
ever, they included both sitting and standing repositioning tasks, mixed different methods of
measuring proprioception–such as active repositioning, or threshold to detection of passive
motion, pooled together different planes of testing, merged lumbo-sacral and trunk proprio-
ception testing, and excluded studies based on reporting methods and data availability [11–
13]. The latest systematic search in previous reviews was conducted in 2014 [12] and it can be
assumed that new evidence on the field has since emerged. From a clinical utility perspective,
none of the previous systematic reviews [11–13] linked the magnitude of the observed proprio-
ceptive deficit to the minimal clinically important difference (MCID). Finally, the vast majority
of studies included in these previous reviews have been cross-sectional, and the reviews
highlighted specific concerns regarding bias. Hence, the main objective of this systematic
review was to evaluate if patients with NSLBP present greater active lumbo-pelvic RE with ref-
erence to seated tasks in the sagittal or transverse plane. Given that mounting evidence sug-
gests that NSLBP is not a homogenous group, but rather denotes a variety of patient
presentations [5, 6], a secondary objective was to evaluate proprioceptive acuity in NSLBP
subgroups.

## Methods

### Protocol and guidelines

The search strategy and reporting of this systematic review adhered to the PRISMA guidelines
[23] and followed the Cochrane group's recommendations [24]. The protocol was prospec-
tively registered in PROSPERO (CRD42018107671).

### Information sources and search methods

PubMed, Cochrane, CINAHL, EMBASE and Web of Science databases were all independently
searched by two reviewers (VK and YS) from inception of database to 28 March 2020 without
language restriction, to reduce language and publication bias.

Grey literature was searched via OpenGrey, and the following registries: Clinical Trials.gov
and EU clinical trials register. Reference lists, citation tracking results, and systematic reviews
were also manually searched.

The search strategy included two basic strings of key terms (low back pain and propriocep-
tion) (S2 File) and followed previously described methodology [11, 19].

### Eligibility criteria

**Types of studies and participants.** The inclusion criteria were a) studies published in
peer-review journals or theses, b) investigating local lumbo-pelvic proprioception between
patients suffering from NSLBP and asymptomatic individuals or matched controls, and c)
measuring proprioception as active RE using a sitting posture as target. Systematic reviews,
case series, case studies, and conference abstracts were excluded, while intervention studies
were retained, but only baseline comparisons were used for data synthesis. Studies were also
excluded if a measure of proprioception was not reported. Participants were considered to suf-
fer NSLBP, if described as such in inclusion criteria, or if serious or specific spinal pathologies,
such as spinal stenosis, or calcification of connective tissue in ankylosing spondylitis, were

used as exclusion criteria [11, 19]. We set no limitations for sex, age, and duration of symptoms of participants.

**Types of outcomes.** At least one outcome measure reflecting active repositioning accuracy, precision and error was the basic eligibility criterion. The following RE indices were selected as appropriate: absolute error (AE) that reflects accuracy or error magnitude, constant error (CE) as an index of bias representing error direction and variable error (VE) representing the variability of an individual's CE [25].

## Study selection

Search results were imported into EndNote and following removal of duplicates, a two-stage screening process was implemented to select relevant studies. Initially, title and abstract were independently evaluated by two reviewers (VK and YS) (minimize selection bias). Subsequently, the full text for each potentially eligible study was evaluated against the criteria for eligibility. A third reviewer (KO) was consulted if consensus was not reached [26].

## Data extraction

A pair of reviewers independently extracted data to enhance transparency (VK and AK). Additionally, pilot testing was performed [24] and the reviewers assessed, practiced and extracted the available data from 30% of the studies [26, 27]. Review authors were not blinded to authors and sources. All data describing study characteristics such as age, sex, sample size, testing procedures, protocols, tasks, variables, and results were obtained and presented.

## Quality assessment

The quality of eligible studies was assessed using a modified Downs and Black's checklist [28]. Each item was scored as "yes", "partially", "no", "unable to determine", or "non-applicable". An overall score was calculated excluding items that were rated as "non-applicable". Following a consensus meeting, the authors judged that twelve items from the checklist as "non-applicable" due to the case–control and cross-sectional design of studies included in this systematic review. Two reviewers (VK and AK) independently rated the quality of included studies, while discrepancies were resolved by discussion with a third reviewer (KO). No studies were excluded due to methodological quality.

## Risk of bias

The risk of bias was assessed at a study level by using an adapted tool suggested by the Non-Randomized Studies Group of the Cochrane Collaboration for systematic reviews of cross-sectional and case–control studies [29, 30]. The following dimensions have been suggested to categorize the risk of bias in non-randomized studies: selection bias, performance bias, detection bias, attrition bias and reporting bias [31]. However, items evaluating performance bias (typically associated with intervention based research) and reporting bias (difficult to quantify [32]) were removed from this tool [29, 30]. Selection bias and control of confounding were evaluated by assessing: a) the appropriate description of characteristics of the participants with NSLBP (i.e., specific inclusion criteria, duration of symptoms, questionnaires evaluating disability) and asymptomatic individuals (i.e., no history of NSLBP pain for "x" weeks/months, no limitations in function), b) the adequacy of the proprioception measurement (i.e. device, apparatus) and the reported reliability (or provided reference) of testing device, c) the validity of the assessment methods of the outcome measures (described in sufficient detail), and d) the adequacy of statistical tests used (description of tests according to data normality, or adjusting

for confounding). Detection bias assessment was based on blinded data assessment or processing (blinded to groups evaluated i.e., NSLBP or asymptomatic), Attrition bias was evaluated from the percentage of the available data for analyses from the recruited participants (<80%). Finally, we evaluated external validity based on the adequate description of participant demographic (i.e., age), and the representativeness of both asymptomatic and NSLBP populations.

## Data analysis, synthesis, and summary of findings

For between-group differences a standardized mean difference (SMD) and 95% confidence interval (95% CI) was calculated to determine the magnitude of difference in RE. When data was available from more than one study, SMDs were pooled in a meta-analysis using random effects (Comprehensive Meta-Analysis software), assuming that the true effect may vary from study to study due to methodological differences (i.e., sample characteristics, reliability of measurement method etc.). SMDs were interpreted as follows: small effect: 0.2–0.6, medium effect: 0.6–1.2, large effect: 1.2–2.0 and very large effect: >2.0 [33].

The $I^2$ statistic was calculated for the evaluation of heterogeneity (also in assessment of inconsistency for the evaluation of the body of evidence). However, heterogeneity was not judged only by the value of $I^2$ statistic, as thresholds for the interpretation can be misleading [34, 35]. First, we assessed the statistical significance of heterogeneity from Q statistic (and *df*), the between-study variance ($Tau^2$), and the distribution (Tau) of the effect sizes about the mean effect (true heterogeneity). Subsequently, we performed visual inspection of the forest plot and the overlap of confidence intervals [24, 34]. Also, given that the $I^2$ statistic provides the proportion of the observed variance that can be attributed to the variance in true effects rather than to sampling error, we also calculated and depicted in the forest plots the prediction interval (±1.96 standard deviations) to evaluate the true effect size range in the meta-analyses [34, 36].

When a study presented only subgroup data, the mean and variance of the composite within a study were computed by performing a fixed-effect meta-analysis on the subgroups for that study. Then, we performed a meta-analysis working solely with these study-level summary effect sizes and variances [37].

Given that subgrouping of NSLBP can reveal characteristics and deficits that were not evident within a broad and heterogeneous NSLBP group [38], quantitative syntheses were grouped according to i) proprioception measurement plane (sagittal or transverse), ii) indices of RE presented (i.e., AE, CE, VE), and iii) according to NSLBP subgroups (FP, AEP, or PEP), the age of the participants (adults or adolescents) and disability (i.e., severity of NSLBP based on disability reported in patient-rated outcome measures), where applicable.

To our knowledge, cut-off values for patient-rated outcome measures have been evaluated in the literature for the Oswestry Disability Index (ODI) [39, 40], but not for the Roland-Morris Disability questionnaire (RMDQ) [41]. Thus, we arbitrarily sub-grouped the included studies based on ODI and RMDQ mean scores (8–15 into mild NSLBP and ≥15 into moderate to severe NSLBP, and scores ≤5 into mild NSLBP and >5 into moderate to severe NSLPB, respectively). For one study, involving two papers [42, 43], which reported no disability score, we categorized the participants as having 'moderate to severe' disability based on 90% reporting back pain requiring medical care or work absenteeism.

Results were presented as summary tables and forest plots. Assessment of publication bias was not possible due to the small number of included trials [44].

Thresholds for clinical interpretation and inferences of the effect size (SMD) of RE have not been established and statistical "rules of thumb" can be misleading. We noted that studies assessing active position sense in both patients and healthy individuals rarely provide

reliability estimates of the study specific measures [16, 42, 43, 45]. However, even a relatively high reliability estimate, such as intraclass correlation coefficient, may not reflect an acceptable measurement if the precision of measurement, as indicated by the standard error of measurement (SEM), is not acceptable for the intended purpose [46]. In practice, the minimum clinically important difference (MCID) for RE would at least exceed the sum of the "noise" in the measurement [46, 47]. Several factors could contribute to the measurement error in lumbo-pelvic proprioceptive acuity, including but not limited to the accuracy of the device utilized, the between-study tester variability, as well as subjects' variation regarding the sitting postures. Based on the eligible studies which reported SEM values, we calculated (smallest real difference = 1.96 x $\sqrt{2}$ x SEM) [48] the mean "noise" of the RE measurement to be 5.4˚, thus we arbitrarily set the MCID at 5˚.

With regards to sensitivity analyses we aimed to repeat the meta-analyses by excluding studies with poor quality and studies appearing as outliers, as well as to present the 95% CIs for the prediction intervals for all pooled effect estimates.

### Assessment of the quality of the body of evidence

Two independent reviewers (VK and AK) evaluated the certainty of evidence using the GRADE methodology [49]. Evidence was started at low certainty, due to predominant case-control and cross-sectional study designs, and was upgraded following published guidelines [50]. Evidence was downgraded according to the presence and extent of four specific factors [51]: (i) high risk of bias (quality appraisal average <60%); (ii) inconsistency of the effect (substantial heterogeneity–$I^2$ > 50%; or large differences in treatment effect estimates, or in the direction of effect across studies); (iii) indirectness (NSLBP patients and/or asymptomatic participants recruited limits generalizability); and (iv) imprecision (upper or lower 95%CI spanned an effect size of 0.5 in either direction; or sample size <400 as "rule of thumb") [29, 30, 52]. We a priori graded an outcome with only one trial as low quality, and if it also had high risk of bias the evidence was graded as very low quality [53].

## Results

### Study selection, study, and participant characteristics

The search strategy identified 605 unique studies, after duplicate removal. A total of 16 paper involving 15 studies met the inclusion criteria. The exclusion of studies at each stage of the selection process is outlined in Fig 1.

Notable reasons for exclusion from the review were: study design, type of publication, repositioning evaluation being performed in standing, repositioning task not focused on the lumbo-pelvic region, and not implementing an active repositioning task (Fig 1).

Study details, participant characteristics, methods of measuring lumbo-pelvic proprioception, and reliability estimates are presented in Table 1. All 16 included studies were published in English and were performed in 9 countries, the most common being Australia (5 studies), USA and UK (2 studies each). The median number of participants recruited per study was 48 (interquartile range 30.0–123.0) and the sample size ranged from 19 to 292 participants. Baseline demographic characteristics did not differ between asymptomatic individuals and NSLBP groups, but body mass index, or body fat, was greater in the NSLBP group in two studies [9, 54]. Also, one study [43] was described across two papers [42] by the same research group and the studies were combined in quantitative synthesis.

Nine studies included patients with NSLBP for more than 3 months [9, 14–16, 43, 54, 55, 59, 61], two studies included patients with NSLBP over 4-week [57] and 6-week [56] duration, and four studies did not report duration of NSLBP for inclusion [20, 58, 60, 62]. Two studies

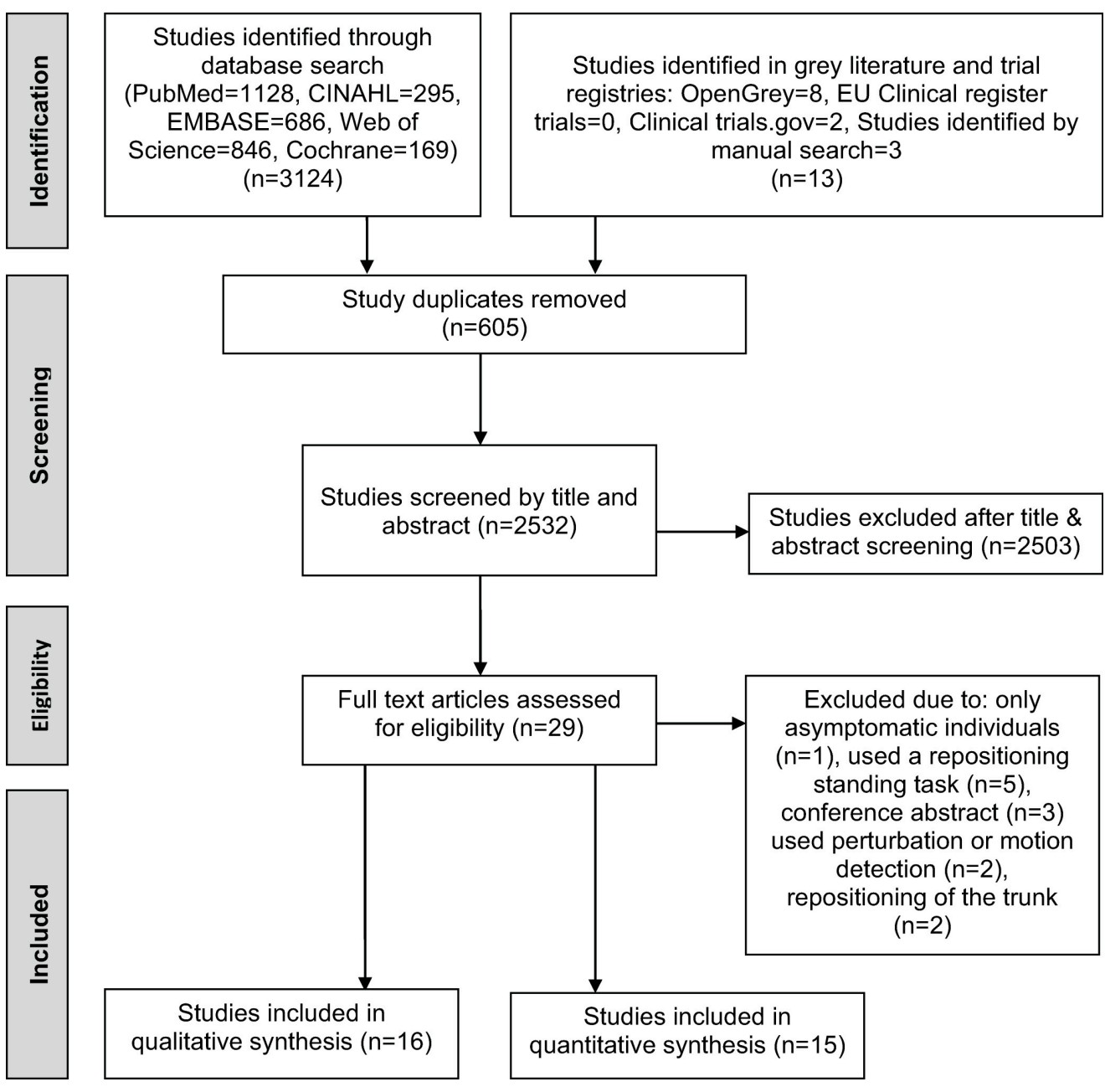

**Fig 1. PRISMA flow diagram for study inclusion.**

included asymptomatic participants that had experienced no NSLBP for at least 2 years [14, 15], five studies included participants with no NSLBP the past year [9, 55–57, 61], three studies recruited participants without a history of NSLBP [16, 54, 59], and five studies did not report specific inclusion criteria except that the participants were asymptomatic the day of testing [20, 42, 58, 60, 62]. Functional disability was reported in all studies except two [43, 58], using either the ODI [14–16, 54, 56, 59–62] or the RMDQ [9, 20, 55–57]. Sub-grouping of patients according to functional movement assessment, testing, or symptomatic severity was conducted in nine studies [9, 14–16, 55–57, 60, 61].

**Table 1. Characteristics of the included studies, participants, and intervention details.**

| Study / design | D & B score (%) | Total sample n* and age** | Disability | Target position and repositioning task | Apparatus, details, and outcome measures | Reliability and SEM |
|---|---|---|---|---|---|---|
| | | | Adult participants–Repositioning task in sagittal plane | | | |
| **Sheeran et al., 2019 [55]** Cross sectional study | 13/16 (82%) | Adults (n = 114) NSLBP (FP = 49) Age: 33.3±10.1 BMI: 25.2 ±3.7 NSLBP (AEP = 23) Age: 39.7±12.9 BMI: 25.0±3.7 NSLBP (PEP = 14) Age: 33.4±8.3 BMI: 25.0±4.5 Controls (n = 28) Age: 35.2±9.7 BMI: 24.8±2.2 | NSLBP (FP) RMDQ: 7.3±3.8 NSLBP (AEP) RMDQ: 6.0±2.8 NSLBP (PEP) RMDQ: 7.1±4.7 | Sagittal plane. Neutral lumbo-pelvic spinal posture after relaxing into usual sitting for 5 seconds. | 4 trials (blindfolded) by using Vicon 512 and Spinal Wheel (Vicon Motion Systems, UK). CE, AE, and VE were reported. NSLBP groups (flexion, passive extension, and active extension) compared to controls. | Not evaluated (referred to reliability from previous study–sitting posture SEM = 1.9–3.7˚). |
| **Korakakis et al., 2019 [56]** Cross sectional study | 14/16 (87.5%) | Adults (n = 30) NSLBP (n = 15) Age: 34.3±9.1 BMI: 23.4±2.1 Controls (n = 15) Age: 33.9±8.7 BMI: 24.5±1.8 | NSLBP RMDQ: 5.1±3.3 ODI: 13.1±7.3 | Sagittal plane. Positioned by therapist in a mid-range and lordotic siting posture (targets). Relaxed in slouch posture and reproduced target postures. | 5 trials (vision available and blind-folded) by using Vicon (Vicon Motion Systems, UK). CE, AE, VE, E and ACE were reported. NSLBP group (MDT extension responder) compared to controls. | Vison available: lordotic posture SEM = 2.46, neutral posture SEM = 1.88. Vison obscured: lordotic posture SEM = 1.45, neutral posture SEM = 1.66. |
| **O'Sullivan et al., 2013 [15]** Cross sectional study | 13/16 (82%) | Adults (n = 30) NSLBP (n = 15) Age: 31.3±10.3 BMI: 24.3 ±3.2 Controls (n = 15) Age: 32.1±9.2 BMI: 23.8±2.0 | NSLBP ODI (%): 14.1±7.8 | Sagittal plane. Neutral lumbo-pelvic spinal posture after adopting a slumped position for 5 seconds. | 3 trials (vision available) by using BodyGuard (Sels Instruments, Belgium). CE, AE, and VE were reported. NSLBP compared to controls. | Not evaluated (referred to reliability from previous study–usual sitting % of flexion ROM, SEM% = 4.78 & 9.03). |
| **Sheeran et al., 2012 [9]** Cross sectional study | 14/16 (87.5%) | Adults (n = 125) NSLBP (FP = 51) Age: 33.0±10.3 BMI: 25.1 ±3.6 NSLBP (AEP = 39) Age: 37.0±11.4 BMI: 24.9±3.8 Controls (n = 15) Age: 36.0±10.3 BMI: 23.3 ±2.2 | NSLBP (FP) RMDQ: 7.3±3.8 NSLBP (AEP) RMDQ: 6.2±3.5 | Sagittal plane. Neutral lumbo-pelvic spinal posture after relaxing into usual sitting for 5 seconds. | 4 trials (blindfolded) by using Vicon 512 and Spinal Wheel (Vicon Motion Systems, UK). CE, AE, and VE were reported. NSLBP groups (Flexion pattern and active extension pattern) compared to controls. | Not evaluated (referred to reliability from previous study–sitting posture SEM = 1.9–3.7˚). |
| **Enoch et al., 2011 [58]** Test-retest reliability study | 10/16 (62.5%) | Adults (n = 40) NSLBP (n = 25) Age: 47.0±12.0 BMI: not reported Controls (n = 15) Age: 45.0±19.0 BMI: not reported | Not reported | Sagittal plane. Neutral lumbo-pelvic posture after moving the pelvis from maximum anterior to maximum posterior tilt. | 3 trials (no info on vision) 5 cm tape-measure with mm markings and a laser pointer. AE was retrieved after request. NSLBP compared to controls. | Not evaluated. However, this was a reliability study reporting test-retest ICC = 0.90, MDC = 0.24. |

*(Continued)*

**Table 1.** (Continued)

| Study / design | D & B score (%) | Total sample n* and age** | Disability | Target position and repositioning task | Apparatus, details, and outcome measures | Reliability and SEM |
|---|---|---|---|---|---|---|
| **Mitchell et al., 2009** [60] Cross sectional study | 15/16 (94%) | Adults (n = 170) NSLBP-Mild (n = 81) Age: 22.0±4.2 f BMI: 23.3±4.3 NSLBP-Significant (n = 53) Age: 23.9±5.1 f BMI: 23.1±3.4 Controls (n = 15) Age: 21.7±3.5 f BMI: 21.9±2.8 | NSLBP-Mild ODI $_{(\%)}$: 10.4±6.6 NSLBP-Significant ODI $_{(\%)}$: 21.2±9.2 | Sagittal plane. Neutral lumbo-pelvic spinal posture after adopting a slumped position for 5 seconds. | 5 trials (vision available) by using Fastrak (Polhemus Navigation Sciences Division, VT). AE was reported. NSLBP groups (mild & significant) compared to controls. | Inter-trial reliability (3 trials for each) mean ICC was 0.97(range: 0.93–0.99) and SEM = 2.0˚ (0.5–2.5˚). |
| **Asell et al., 2006** [61] Cross sectional study | 13/16 (82%) | Adults (n = 123) CNSLBP (n = 92) Age: 38.0±7.0 BMI: not reported Controls (n = 31) Age: 36.0±9.0 BMI: not reported | CNSLBP ODI $_{(\%)}$: 29.0±12.0 | Sagittal plane. The target position was set 1/3 of the way towards maximal extension from the subjects' normal sitting position. | 5 trials (blindfolded) by using Fastrak (Polhemus Navigation Sciences Division, VT). VE and CE were reported. NSLBP groups compared to controls. | ICC with absolute agreement showed an ICC$_{VE}$ = 0.754 and an ICC$_{CE}$ = 0.860. |
| **O'Sullivan et al., 2003** [14] Cross sectional study | 12/16 (75%) | Adults (n = 30) CNSLBP (FP = 15) Age: 38.8±12.0 BMI: 25.2 Controls (n = 15) Age: 38.2±10.9 BMI: 24.1 | CNSLBP ODI $_{(\%)}$:26.1±13.3 | Sagittal plane. Neutral lumbo-pelvic spinal posture after adopting full lumbar flexion for 5 seconds. | 5 trials (blindfolded) by using Fastrak (Polhemus Navigation Sciences Division, VT). AE was reported. NSLBP groups compared to controls. | Not evaluated (referred to reliability from previous study–standing max ROM, error <0.2˚). |
| **Brumagne et al., 2000** [62] Cross sectional study | 11/16 (69%) | Adults (n = 44) NSLBP (n = 15) Age: 21.8±2.1 BMI: 21.5 Controls (n = 15) Age: 22.3±3.8 BMI: 20.9 | NSLBP ODI $_{(\%)}$:7.0±6.8 | Sagittal plane. Neutral lumbar sitting posture after full anterior pelvic tilt. | 5 trials (blindfolded) by using piezoresistive electro-goniometer (Eurosensors, UK). CE, AE, and VE were reported. NSLBP group compared to controls. | Not evaluated (referred to reliability from previous study–range of pelvic tilting, accuracy 0.42˚). |
| **Haydu et al., 2000** [54] Cross sectional study | 13/16 (82%) | Adults (n = 19) CNSLBP (n = 9) Age: 44.4±5.27 BMI: 31.0±8.92 Controls (n = 10) Age: 44.2±8.3 BMI: 27.1±3.98 | CNSLBP ODI $_{(\%)}$:23.8±26.2 | Sagittal plane. Neutral lumbar sitting posture after extreme posterior and anterior pelvic tilt. | 10 trials (vision available) electronic goniometer (elgon). AE was reported. CNSLBP group compared to controls. | Not evaluated, (referred to reliability from previous studies in standing–ICC 0.84–097). |
| **Lam et al., 1999** [43] Single group cross sectional study | 8/11 (73%) | Adults (n = 20) NSLBP Age: 29.0±5.0 (19–36) BMI: not reported | Not reported | Sagittal plane. Reposition to neutral lumbar sitting posture after relaxed full lumbar flexion. | 3 trials (blindfolded) by using Fastrak (Polhemus Navigation Sciences Division, VT). AE only reported. | Not evaluated (referred to reliability from previous study–lumbar spine in standing, accuracy 0.2˚). |
| **Maffey-Ward et al, 1996** [42] Single group cross sectional study | 8/11 (73%) | Adults (n = 10) Controls Age: 29.0 (19–34) BMI: not reported | Not applicable | Sagittal plane. Reposition to neutral lumbar sitting posture after relaxed full lumbar flexion | 3 trials (blindfolded) by using Fastrak (Polhemus Navigation Sciences Division, VT). Root mean square AE was reported. | Not evaluated (referred to reliability from previous study–lumbar spine in standing, accuracy 0.2˚). |
| **Adolescent participants—Repositioning task in sagittal plane** | | | | | | |

(*Continued*)

**Table 1.** (Continued)

| Study / design | D & B score (%) | Total sample n* and age** | Disability | Target position and repositioning task | Apparatus, details, and outcome measures | Reliability and SEM |
|---|---|---|---|---|---|---|
| **Astfalck et al., 2013** [16] Cross sectional study | 15/16 (94%) | Adolescents (n = 56) NSLBP (n = 28) Age:15.4±0.5 BMI: 22.2 ±3.5 NSLBP (FP = 15) Age: 15.4±0.5 BMI: 21.6±2.8 NSLBP (AEP = 13) Age: 15.4±0.6 BMI: 22.8±4.2 Controls (n = 28) Age: 15.7±0.5 BMI: 21.2±2.6 | NSLBP (n = 28) ODI $_{(\%)}$:17.9±10.1 NSLBP (FP) ODI $_{(\%)}$:19.0±11.9 NSLBP (FP) ODI $_{(\%)}$:19.0±11.9 | Sagittal plane. Positioned by therapist in a mid-range siting posture (target). Relaxed in slouch posture and reproduced target posture. | 3 trials (blindfolded) by using Fastrak (Polhemus Navigation Sciences Division, VT). CE, AE, and VE were reported. Three groups: flexion pattern, multidirectional and controls. | Not evaluated (referred to reliability from previous study–standing max ROM, error <0.2˚). |
| **Adult participants—Repositioning task in transverse plane** ||||||| 
| **Boucher et al., 2017** [57] Non-randomized controlled study | 13/16 (82%) | Adults (n = 60) NSLBP (n = 40) Age: 38.6±10.3 m BMI: 25.0±2.9 m Age: 46.3±11.2 f BMI: 24.4±2.9 f Controls (n = 20) Age: 39.7±13.8 m BMI: 24.8±2.2 m Age: 39.8±13.5 f BMI: 22.9±2.8 | NSLBP RMDQ: 6.4±3.7 m RMDQ: 3.6±2.5 f | Transverse plane. 10˚ of axial rotation (left or right) used as target position, relative to the participants' neutral (zero). | 10 trials (blindfolded) by using custom-built apparatus. CE, AE, VE, and E were reported. NSLBP compared to controls. | Not evaluated (referred to reliability from previous study–perception of motion SEM = 1.2˚). |
| **Lee et al., 2010** [59] Cross sectional study | 11/16 (69%) | Adults (n = 48) NSLBP (n = 24) Age:42.6±13.7 BMI: 24.96 Controls (n = 24) Age: 42.4±9.0 BMI: 24.96 | NSLBP ODI $_{(\%)}$: 19.0±15.0 | Transverse plane. 15˚ of axial rotation and patients were aiming as target position a neutral sitting posture. | 4 trials (blindfolded) by using custom-built apparatus in each direction. AE was reported. NSLBP group compared to controls. | Not evaluated. The resolution of the apparatus was less than 0.01˚, and the accuracy of the calibration curve was 0.35˚. |
| **Silfies et al., 2007** [20] Prospective cohort study | 13/16 (82%) | Adults (n = 303) NSLBP (n = 40) Age: 19.8±1.2 BMI: not available Controls (n = 263) Age: 19.3±2.5 BMI: not available | Not available | Transverse plane. 20˚ of axial rotation and patients were aiming as target position a neutral sitting posture. | 8 trials (blindfolded) by using custom-built apparatus. AE and VE were reported. NSLBP group compared to controls. | AE active repositioning SEM = 0.57˚ VE active repositioning SEM = 0.58˚. |

*n = number of individuals in each group.

**Age in mean ± SD or mean (range) as reported for each group in the study.

Abbreviations: D & B score, Downs and Black checklist score (higher score indicates higher study quality); RMDQ, Roland Morris Disability Questionnaire; ODI, Oswestry Disability Index; m, male; f, female; SEM, standard error of measurement; NSLBP, non-specific low back pain; AE, absolute error, CE, constant error; VE, variable error, E, total variability, ACE, absolute constant error; ROM, range of motion; FP, flexion pattern, AEP, active extension pattern; ICC, intraclass correlation coefficient; PEP, passive extension pattern.

A neutral lumbo-pelvic spinal posture was used by 15 studies as a target sitting posture, one study used in addition a second (lordotic) target posture, while in one study [61] the target position was set at 1/3 of the way towards maximal extension from the subjects' normal sitting position. Participants were blindfolded for testing in 12 studies, four studies [15, 54, 56, 60] evaluated repositioning accuracy with vision available, while one study [58] did not provide relevant information (Table 1).

## Quality assessment

The quality rating scores (Table 1) on the checklist ranged from 62.5% to 94% (median = 82, interquartile range: 73–86.1). The quality assessment indicated that all studies had clear hypotheses and objectives, clearly described participants' demographics, quality of outcome measures and description of results. In contrast, common concerns included 87.5% of studies lacking a sample size calculation, 56% of studies not adjusting for confounders, and 50% not providing adequate sampling information.

## Risk of bias

All studies had a high risk of bias for detection (assessment or processing of data by a blinded assessor), five studies failed to adequately describe the population of interest, six studies failed to report adequately eligibility criteria, and two studies did not apply relevant statistical analysis (Table 2).

**Absolute repositioning error–sagittal plane.** Twelve studies [9, 14–16, 42, 43, 54–56, 58, 60, 62] evaluated AE among NSLBP patients compared to asymptomatic individuals in a sagittal plane repositioning task.

**Absolute error between asymptomatic individuals and NSLBP patients.** Pooled results revealed a medium effect for greater AE in patients with NSLBP than asymptomatic individuals (SMD = 0.705, 95%CI: 0.199–1.212) (Fig 2A). Removing three studies [14, 15, 58] that did not report AE in angular measures had a significant impact on the direction of the effect

**Table 2. Risk of bias assessment.**

| Study | External validity | | | Internal validity | | | | Score (%) |
|---|---|---|---|---|---|---|---|---|
| | | Detection | Attrition | Selection bias / control of confounding | | | | |
| | a | b | c | d | e | f | g | |
| Sheeran et al, 2019 | Yes | No | Yes | No | Yes | Yes | Yes | 71.4 |
| Korakakis et al, 2019 | Yes | No | Yes | Yes | Yes | Yes | Yes | 85.7 |
| Boucher et al, 2017 | Yes | No | Yes | Yes | Yes | Yes | Yes | 85.7 |
| Asfalck et al, 2013 | Yes | No | Yes | Yes | Yes | Yes | Yes | 85.7 |
| O'Sullivan et al, 2013 | Yes | No | Yes | Yes | Yes | Yes | Yes | 85.7 |
| Sheeran et al, 2012 | Yes | No | Yes | No | Yes | Yes | Yes | 71.4 |
| Enoch et al, 2011 | Yes | No | Yes | No | Yes | Yes | No | 57.1 |
| Lee et al, 2010 | Yes | No | Yes | Yes | Yes | Yes | Yes | 85.7 |
| Mitchell et al, 2009 | No | No | Yes | Yes | Yes | Yes | Yes | 85.7 |
| Silfies et al, 2007 | No | No | Yes | No | Yes | Yes | Yes | 57.1 |
| Asell et al, 2006 | Yes | No | Yes | Yes | Yes | Yes | Yes | 85.7 |
| O'Sullivan et al, 2003 | Yes | No | Yes | Yes | Yes | Yes | Yes | 85.7 |
| Brumagne et al, 2000 | No | No | Yes | No | Yes | Yes | Yes | 57.1 |
| Haydu, 2000 | No | No | Yes | Yes | Yes | Yes | Yes | 71.4 |
| Lam, 1999 & Maffey-Ward, 1996 | No | No | Yes | No | Yes | Yes | No | 42.8 |

*Notes*: a) Representative: if the study described demographic details (i.e. age and sex), and were representative of both asymptomatic and NSLBP population, b) Blinded assessor: if data assessed or processed by a blinded assessor (blinded to groups evaluated i.e. NSLBP or asymptomatic), c) Attrition: if >80% of data was available for analyses from those recruited, d) Appropriate description of characteristics of the participants with NSLBP (i.e. specific inclusion criteria, duration of symptoms, questionnaires evaluating pain and function) and asymptomatic individuals (i.e. no history of NSLBP pain for "x" weeks/months, no limitations in function, no pain), e) Appropriate proprioception measurement (i.e. device, apparatus) and reported reliability (or provided reference) of testing device, f) Appropriate methods of assessment of outcome measures (described in sufficient detail), g) Appropriate statistical tests used: described type of tests (i.e. parametric, non-parametric) according to normality, or adjusting for confounding.

**a**

| Study name | SMD | SE | Variance | 95% CI | | Z-Value | p-Value | LBP | Control | Std diff in means and 95% CI Random effects | Relative weight |
|---|---|---|---|---|---|---|---|---|---|---|---|
| **Absolute Error - Sagittal plane** | | | | | | | | | | | |
| Lam,1999 & Maffey-Ward,1996 | -0.352 | 0.390 | 0.152 | -1.116 | 0.412 | -0.903 | 0.367 | 20 | 10 | | 8.55 |
| Brumagne et al, 2000 | 0.642 | 0.309 | 0.096 | 0.036 | 1.249 | 2.076 | 0.038 | 23 | 21 | | 9.21 |
| Haydu, 2000 | -0.255 | 0.461 | 0.213 | -1.160 | 0.649 | -0.554 | 0.580 | 9 | 10 | | 7.93 |
| O'Sullivan et al, 2003 | 0.849 | 0.381 | 0.145 | 0.101 | 1.596 | 2.226 | 0.026 | 15 | 15 | | 8.62 |
| Mitchell et al, 2009 | 0.147 | 0.148 | 0.022 | -0.144 | 0.438 | 0.991 | 0.322 | 134 | 36 | | 10.25 |
| Enoch et al, 2010 | 0.463 | 0.331 | 0.109 | -0.185 | 1.111 | 1.399 | 0.162 | 25 | 15 | | 9.04 |
| Sheeran et al, 2012 | 1.680 | 0.226 | 0.051 | 1.237 | 2.122 | 7.440 | 0.000 | 90 | 35 | | 9.82 |
| Astfalck et al, 2013 | 0.535 | 0.272 | 0.074 | 0.002 | 1.068 | 1.968 | 0.049 | 28 | 28 | | 9.49 |
| O'Sullivan et al, 2013 | 1.233 | 0.398 | 0.159 | 0.452 | 2.013 | 3.095 | 0.002 | 15 | 15 | | 8.47 |
| Korakakis et al, 2019 | 0.426 | 0.369 | 0.136 | -0.297 | 1.150 | 1.155 | 0.248 | 15 | 15 | | 8.72 |
| Sheeran et al, 2019 | 2.113 | 0.212 | 0.045 | 1.697 | 2.529 | 9.961 | 0.000 | 86 | 28 | | 9.90 |
| **Total** | **0.705** | **0.258** | **0.067** | **0.199** | **1.212** | **2.730** | **0.006** | | | | |

Heterogeneity: Tau² = 0.629; Q = 91.206, df = 10 (p < 0.000); I² = 89.036%
Test for overall effect: Z = 2.730 (p = 0.006)

**b**

| Study name | SMD | SE | Variance | 95% CI | | Z-Value | p-Value | LBP | Control | Std diff in means and 95% CI Random effects | Relative weight |
|---|---|---|---|---|---|---|---|---|---|---|---|
| **Absolute error moderate to severe LBP - Sagittal plane** | | | | | | | | | | | |
| Lam,1999 & Maffey-Ward,1996 | -0.352 | 0.390 | 0.152 | -1.116 | 0.412 | -0.903 | 0.367 | 20 | 10 | | 13.54 |
| Haydu, 2000 | -0.255 | 0.461 | 0.213 | -1.160 | 0.649 | -0.554 | 0.580 | 9 | 10 | | 12.76 |
| O'Sullivan et al, 2003 | 0.849 | 0.381 | 0.145 | 0.101 | 1.596 | 2.226 | 0.026 | 15 | 15 | | 13.63 |
| Mitchell, 2009 | 0.245 | 0.217 | 0.047 | -0.179 | 0.670 | 1.132 | 0.258 | 53 | 36 | | 15.14 |
| Sheeran et al, 2012 | 1.680 | 0.226 | 0.051 | 1.237 | 2.122 | 7.440 | 0.000 | 90 | 35 | | 15.07 |
| Astfalck et al, 2013 | 0.535 | 0.272 | 0.074 | 0.002 | 1.068 | 1.968 | 0.049 | 28 | 28 | | 14.69 |
| Sheeran et al, 2019 | 2.113 | 0.212 | 0.045 | 1.697 | 2.529 | 9.961 | 0.000 | 86 | 28 | | 15.17 |
| **Total** | **0.725** | **0.366** | **0.134** | **0.007** | **1.443** | **1.978** | **0.048** | | | | |

Heterogeneity: Tau² = 0.840; Q = 71.213, df = 6 (p < 0.000); I² = 91.575%
Test for overall effect: Z = 1.978 (p = 0.048)

**c**

| Study name | SMD | SE | Variance | 95% CI | | Z-Value | p-Value | LBP | Control | Std diff in means and 95% CI Random effects | Relative weight |
|---|---|---|---|---|---|---|---|---|---|---|---|
| **Absolute error mild LBP - Sagittal plane** | | | | | | | | | | | |
| Brumagne et al, 2000 | 0.642 | 0.309 | 0.096 | 0.036 | 1.249 | 2.076 | 0.038 | 23 | 21 | | 25.33 |
| Mitchell et al, 2009 | 0.064 | 0.200 | 0.040 | -0.329 | 0.457 | 0.319 | 0.750 | 81 | 36 | | 32.78 |
| O'Sullivan et al, 2013 | 1.233 | 0.398 | 0.159 | 0.452 | 2.013 | 3.095 | 0.002 | 15 | 15 | | 20.16 |
| Korakakis et al, 2019 | 0.426 | 0.369 | 0.136 | -0.297 | 1.150 | 1.155 | 0.248 | 15 | 15 | | 21.73 |
| **Total** | **0.525** | **0.249** | **0.062** | **0.037** | **1.013** | **2.107** | **0.035** | | | | |

Heterogeneity: Tau² = 0.149; Q = 7.799, df = 3 (p = 0.050); I² = 61.533%
Test for overall effect: Z = 2.107 (p = 0.035)

**d**

| Study name | SMD | SE | Variance | 95% CI | | Z-Value | p-Value | LBP | Control | Std diff in means and 95% CI Random effects | Relative weight |
|---|---|---|---|---|---|---|---|---|---|---|---|
| **Absolute error in flexion aggravated LBP - Sagittal plane** | | | | | | | | | | | |
| O'Sullivan et al, 2003 | 0.849 | 0.381 | 0.145 | 0.101 | 1.596 | 2.226 | 0.026 | 15 | 25 | | 15.41 |
| Sheeran et al, 2012 | 1.950 | 0.265 | 0.070 | 1.431 | 2.470 | 7.356 | 0.000 | 51 | 35 | | 18.74 |
| Astfalck et al, 2013 | 0.855 | 0.333 | 0.111 | 0.202 | 1.507 | 2.567 | 0.010 | 15 | 28 | | 16.78 |
| O'Sullivan et al, 2013 | 1.233 | 0.398 | 0.159 | 0.452 | 2.013 | 3.095 | 0.002 | 15 | 15 | | 14.94 |
| Korakakis et al, 2019 | 0.426 | 0.369 | 0.136 | -0.297 | 1.150 | 1.155 | 0.248 | 15 | 15 | | 15.74 |
| Sheeran et al, 2019 | 1.784 | 0.277 | 0.077 | 1.241 | 2.327 | 6.438 | 0.000 | 49 | 28 | | 18.39 |
| **Total** | **1.219** | **0.255** | **0.065** | **0.719** | **1.719** | **4.778** | **0.000** | | | | |

Heterogeneity: Tau² = 0.277; Q = 17.838, df = 5 (p = 0.003); I² = 71.970%
Test for overall effect: Z = 4.778 (p = 0.000)

**e**

| Study name | SMD | SE | Variance | 95% CI | | Z-Value | p-Value | LBP | Control | Std diff in means and 95% CI Random effects | Relative weight |
|---|---|---|---|---|---|---|---|---|---|---|---|
| **Absolute error in extension aggravated LBP - Sagittal plane** | | | | | | | | | | | |
| Sheeran et al, 2012 | 1.871 | 0.279 | 0.078 | 1.324 | 2.418 | 6.704 | 0.000 | 39 | 35 | | 34.19 |
| Astfalck et al, 2013 | 0.194 | 0.336 | 0.113 | -0.466 | 0.853 | 0.576 | 0.565 | 13 | 28 | | 33.20 |
| Sheeran et al, 2019 | 2.398 | 0.368 | 0.136 | 1.676 | 3.120 | 6.513 | 0.000 | 23 | 28 | | 32.60 |
| **Total** | **1.486** | **0.636** | **0.405** | **0.239** | **2.732** | **2.336** | **0.019** | | | | |

Heterogeneity: Tau² = 1.105; Q = 22.803, df = 2 (p = 0.000); I² = 91.229%
Test for overall effect: Z = 2.336 (p = 0.019)

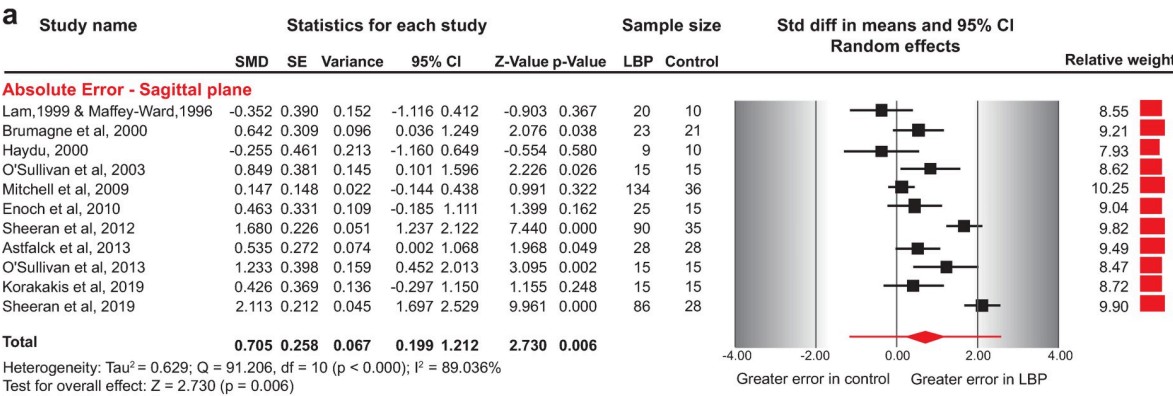

**Fig 2. Forest plots for absolute error in sagittal plane repositioning between NSLBP and asymptomatic individuals.** a) Absolute error between asymptomatic individuals and NSLBP patients, b) Absolute error between asymptomatic individuals and patients with moderate to severe NSLPB symptoms, c) Absolute error between asymptomatic individuals and patients with mild NSLBP symptoms, d) Absolute error between asymptomatic individuals and patients with flexion-aggravated NSLBP, and e) Absolute error between asymptomatic individuals and patients with extension-aggravated NSLBP. Data for one study [58] was requested and provided by the authors, and two papers [42, 43] from the same research group were combined in quantitative synthesis as one served as the NSLBP group and the other as the control group. The line crossing the summary effect estimate (diamond) in the forest plots represents the prediction interval, assuming that the effects are normally distributed [36]. Abbreviations: SMD, standardized mean difference; SE, standard error; CI, confidence intervals; NSLBP, non-specific low back pain.

estimate, showing no significant difference between groups (SMD = 0.651, 95%CI: 0.000–1.302) (Table 3).

## Absolute error and subgroup analyses

Four studies [15, 56, 60, 62] included patients with mild NSLBP symptoms (ODI<15 or RMDQ<5 points). We found a significant effect of symptoms' severity in AE, with patients with moderate to severe NSLBP presenting larger errors (medium effect) (SMD = 0.725, 95%CI: 0.007–1.443) compared to asymptomatic individuals than patients with mild NSLBP symptoms (small effect) (SMD = 0.525, 95%CI: 0.37–1.013) (Fig 2B and 2C, Table 3).

Six studies evaluated AE in specific directional sub-groups of NSLBP patients [9, 14–16, 55, 56]. Five studies used O'Sullivan's classification [5], while one used a syndrome-based classification [56]; however, both categorize patients on the basis of pain-provoking movements and postures. All studies reported data for NSLBP related to postures and movements involving flexion and three studies [9, 16, 55] also included extension-aggravated NSLBP patients.

Pooled results revealed a large effect for greater AE for patients with flexion-aggravated NSLBP than asymptomatic individuals (SMD = 1.219, 95%CI: 0.719–1.719) (Fig 2D). Removing two studies [14, 15] from data synthesis that did not report AE in angular measures or excluding adolescent patients [16] had no significant impact on the direction and the size of the effect estimate (SMD = 1.287, 95%CI: 0.595–1.980; SMD = 1.289, 95%CI: 0.719–1.859, respectively) (Table 3).

The pooled results for extension-aggravated NSLBP revealed a large effect for greater AE for NSLBP patients than asymptomatic individuals (SMD = 1.486, 95%CI: 0.239–2.732) (Fig 2E). Excluding adolescents [16] had a significant impact on the size of the effect estimate, that increased the large effect to a very large effect for greater AE in extension- aggravated NSLBP patients (SMD = 2.080, 95%CI: 1.574–2.585) (Table 3).

One study [55] evaluating patients with NSLBP aggravated by passive extension presented very low certainty evidence of greater AE in patients compared to asymptomatic individuals (MD = 8.8, 95%CI: 7.006–10.594).

In eight studies [9, 14, 16, 42, 43, 55, 56, 62] participants were blindfolded during testing, while in five studies participants had vision available [15, 54, 56, 58, 60]. The pooled results indicated a medium effect for greater AE for NSLBP patients than asymptomatic individuals (SMD = 0.876, 95%CI: 0.231–1.521) for the blindfolded repositioning task, whereas AE in neutral sitting posture did not differ between NSLBP and asymptomatic individuals when vision was available (SMD = 0.325, 95%CI: -0.075 to 0.725) (Fig 3A and 3B, Table 3).

One study [56] evaluating AE in repositioning into a lordotic sitting posture presented very low certainty evidence of no difference between patients with mild NSLBP and asymptomatic individuals with participants blindfolded or not (MD = -1.9, 95%CI: -4.053 to 0.253; MD = -1.2, 95%CI: -3.073 to 0.673, respectively).

**Table 3. Summary of evidence for repositioning error.**

| Outcome | Comparisons | | Relative effect (95% CI) | Patients /controls (studies) | Quality of evidence (GRADE) | Evidence and significance |
|---|---|---|---|---|---|---|
| | Average estimate in NSLBP group | Average estimate in control group | | | | |
| Absolute error | NSLBP: | Control: | SMD 0.705 | 460/228 | ⊕◯◯◯ | Very low-level evidence of greater (medium effect) AE in NSLBP patients than in asymptomatic individuals |
| All studies | Pooled weighted mean±SD not estimable* | Pooled weighted mean±SD not estimable* | [0.199, 1.212] Statistically significant difference | (11) | Very low[1,2,3] | |
| Absolute error | NSLBP: | Control: | SMD 0.651 | 405/183 | ⊕◯◯◯ | Very low-level evidence of no difference in AE between NSLBP patients and asymptomatic individuals |
| Studies reporting AE in degrees | Pooled weighted mean±SD was 4.6±3.2˚ (mean range 0.9 to 10.6) | Pooled weighted mean±SD was 1.8 ±2.0˚ (mean range 0.7 to 4.47) | [0.000, 1.302] Statistically significant difference | (8) | Very low[1,2,3] | |
| Absolute error | NSLBP: | Control: | SMD 0.720 | 432/200 | ⊕◯◯◯ | Very low-level evidence of greater (medium effect) AE in NSLBP patients than in asymptomatic individuals |
| (adults only) | Pooled weighted mean±SD not estimable* | Pooled weighted mean±SD not estimable* | [0.160, 1.281] Statistically significant difference | (10) | Very low[1,2,3] | |
| Absolute error | Mild-NSLBP: | Control: | SMD 0.525 | 134/87 | ⊕⊕◯◯ | Moderate level evidence of greater (small effect) AE in patients with mild NSLBP than in asymptomatic individuals |
| Mild NSLBP | Pooled weighted mean±SD not estimable* | Pooled weighted mean±SD not estimable* | [0.037, 1.013] Statistically significant difference | (4) | Low[2,3] | |
| Absolute error | Moderate-NSLBP: Pooled weighted mean±SD not estimable* | Control: | SMD 0.725 | 301/162 | ⊕◯◯◯ | Very low-level evidence of greater (medium effect) AE in patients with moderate to severe NSLBP than in asymptomatic individuals |
| Moderate to severe NSLBP | | Pooled weighted mean±SD not estimable* | [0.007, 1.443] Statistically significant difference | (7) | Very low[1,2,3] | |
| Absolute error | FP-NSLBP: | Control: | SMD 1.219 | 160/136 | ⊕◯◯◯ | Very low-level evidence of greater (large effect) AE in FP subgroup of NSLBP patients than in asymptomatic individuals |
| FP NSLBP subgroup | Pooled weighted mean±SD not estimable* | Pooled weighted mean±SD not estimable* | [0.719, 1.719] Statistically significant difference | (6) | Very low[1,2,3] | |
| Absolute error | FP-NSLBP: | Control: | SMD 1.287[a] | 130/106 | ⊕◯◯◯ | Very low-level evidence of greater (large effect) AE in FP subgroup of NSLBP patients than in asymptomatic individuals |
| FP NSLBP subgroup (error in degrees) | Pooled weighted mean±SD was 7.1±3.9˚ (mean range 4.6 to 7.9) | Pooled weighted mean±SD was 2.3 ±1.1˚ (mean range 0.7 to 4.47) | [0.595, 1.980] Statistically significant difference | (4) | Very low[1,2,3] | |
| Absolute error | FP-NSLBP: | Control: | SMD 1.289 | 145/108 | ⊕⊕◯◯ | Low level evidence of greater (large effect) AE in FP adult subgroup of NSLBP patients than in asymptomatic individuals |
| FP NSLBP subgroup (adults) | Pooled weighted mean±SD not estimable* | Pooled weighted mean±SD not estimable* | [0.719, 1.859] Statistically significant difference | (5) | Low[2,3] | |
| Absolute error | AEP-NSLBP: Pooled weighted mean±SD was 6.3 ±3.2˚ (mean range 3.4 to 7.6) | Control: | SMD 1.486[b] | 75/91 | ⊕◯◯◯ | Very low-level evidence of greater (large effect) AE in AEP subgroup of NSLBP patients than in asymptomatic individuals |
| AEP NSLBP subgroup | | Pooled weighted mean±SD was 2.2 ±0.9˚ (mean range 1.8 to 3.1) | [0.239, 2.732] Statistically significant difference | (3) | Very low[1,2,3] | |
| Absolute error | AEP-NSLBP: | Control: | SMD 2.080[c] | 62/63 | ⊕⊕⊕◯ | Moderate level evidence of greater (very large effect) AE in AEP adult subgroup of NSLBP patients than in asymptomatic individuals |
| AEP NSLBP subgroup (adults) | Pooled weighted mean±SD was 6.9±3.5˚ (mean range 5.7 to 7.6) | Pooled weighted mean±SD was 1.8 ±0.8˚ (mean range 1.8 to 1.8) | [1.574, 2.585] Statistically significant difference | (2) | Moderate[3] | |

(*Continued*)

**Table 3.** (Continued)

| Outcome | Comparisons | | Relative effect (95% CI) | Patients /controls (studies) | Quality of evidence (GRADE) | Evidence and significance |
|---|---|---|---|---|---|---|
| | Average estimate in NSLBP group | Average estimate in control group | | | | |
| **Absolute error** Vision obscured | NSLBP: Pooled weighted mean±SD not estimable* | Control: Pooled weighted mean±SD not estimable* | **SMD 0.876** [0.231, 1.521] Statistically significant difference | 277/152 (7) | ⊕◯◯◯ **Very low**[1,2,3] | Very low-level evidence of greater (medium effect) AE in blindfolded NSLBP patients than in asymptomatic individuals |
| **Absolute error** Vision available | NSLBP: Pooled weighted mean±SD not estimable* | Control: Pooled weighted mean±SD not estimable* | **SMD 0.325** [-0.075, 0.725] Non-statistically significant difference | 198/91 (5) | ⊕◯◯◯ **Very low**[1,2,3] | Very low-level evidence of no difference in AE between NSLBP patients and asymptomatic individuals with vision available |
| **Absolute error** Transverse plane | NSLBP: Pooled weighted mean±SD was 2.7±0.5° (mean range 1.6 to 4.4) | Control: Pooled weighted mean±SD was 2.5 ±0.7° (mean range 1.6 to 3.6) | **SMD 0.665** [-0.495, 1.825] Non-statistically significant difference | 102/306 (3) | ⊕◯◯◯ **Very low**[1,2,3] | Very low-level evidence of no difference in transverse plane AE between NSLBP patients and asymptomatic individuals |

[1] Downgraded due to indirectness.

[2] Downgraded due to inconsistency.

[3] Downgraded due to imprecision.

* Pooled weighted mean not estimable as included studies did not report AE, VE, or CE as an angular measure, or did not report values for mean±SD.

[a] Pooled weighted SD was 2.64° resulting in an effect estimate of 3.4°—non-clinically significant.

[b] Pooled weighted SD was 1.94° resulting in an effect estimate of 2.9°—non-clinically significant.

[c] Pooled weighted SD was 2.1° resulting in an effect estimate of 4.4°—non-clinically significant.

Abbreviations: SMD, standardized mean difference, CI, confidence intervals; NSLBP, non-specific low back pain; SD, standard deviation; AE, absolute error; NSLBP, non-specific low back pain; FP, flexion pattern; AEP, active extension pattern.

## Absolute repositioning error–transverse plane

Three studies [20, 57, 59] evaluated proprioception among blindfolded NSLBP patients compared to asymptomatic individuals in a transverse plane repositioning task (axial lumbar rotation).

The included studies [20, 57, 59] reported conflicting results; however, when the data were pooled into a summary estimate (Fig 3C), no significant differences were evident between NSLBP patients and asymptomatic individuals in a seated repositioning task involving axial lumbar rotation (SMD = 0.665, 95%CI -0.495 to 1.825) (Table 3).

**Variable repositioning error–sagittal plane.** Seven studies [9, 16, 55, 56, 61, 62] evaluated the variability of repositioning of NSLBP patients compared to asymptomatic individuals in a sagittal plane repositioning task.

**Variable error between asymptomatic individuals and NSLBP patients.** The included studies presented conflicting results. Pooled data into a summary estimate showed that patients with NSLBP had greater variability (medium effect) in RE about their mean response than asymptomatic individuals (SMD = 0.606, 95%CI 0.114–1.098) (Fig 4A). Removing studies that did not report VE in angular measures [15], or that included adolescents [16] did not impact the direction and the size of the effect estimate (SMD = 0.655, 95%CI: 0.111–1.200; SMD = 0.681, 95%CI: 0.135–1.227, respectively) (Table 4).

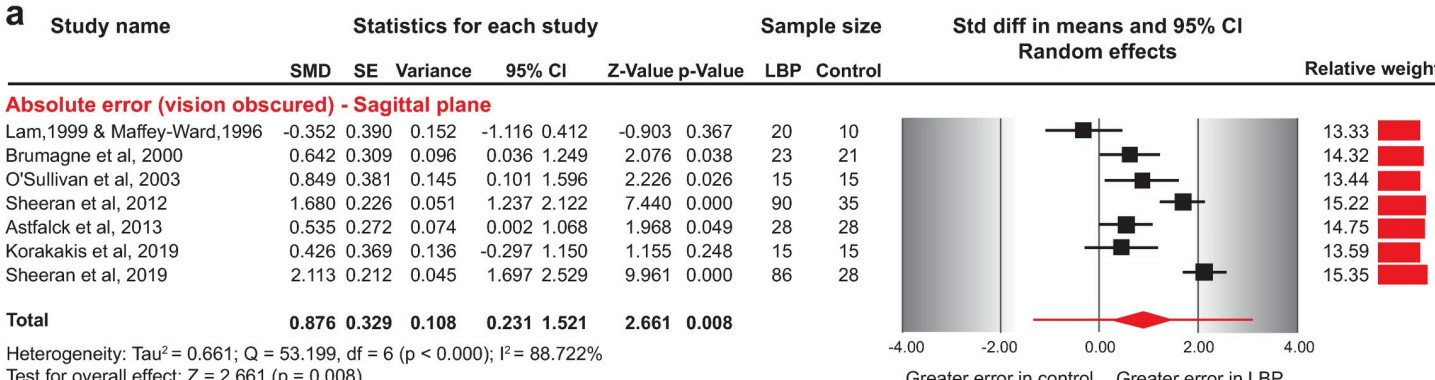

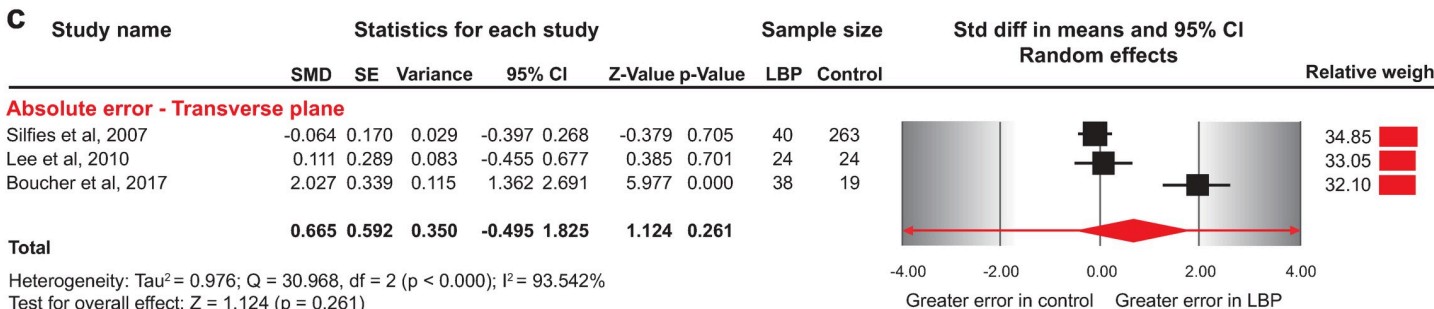

**Fig 3. Forest plots for absolute error in sagittal plane repositioning between NSLBP and asymptomatic individuals.** a) Absolute error between blindfolded asymptomatic individuals and NSLBP patients, b) Absolute error between asymptomatic individuals and NSLBP patients evaluated with vision available, and c) Absolute error between asymptomatic individuals and patients with NSLBP evaluated in transverse plane. Data for two studies [20, 58] was requested and provided by the authors, and two studies [42, 43] from the same research group were combined in quantitative synthesis as one served as the NSLBP group and the other as the control group. The line crossing the summary effect estimate (diamond) in the forest plots represents the prediction interval, assuming that the effects are normally distributed [36]. Abbreviations: SMD, standardized mean difference; SE, standard error; CI, confidence intervals; NSLBP, non-specific low back pain.

## Variable error and subgroup analyses

By subgrouping studies according to NSLBP severity, we did not find a significant effect of symptoms' severity in VE, with both patients with mild and moderate to severe NSLBP presenting no difference compared to asymptomatic individuals (SMD = 0.609, 95%CI: -0.226 to 1.444; SMD = 0.602, 95%CI: -0.071 to 1.275, respectively) (Fig 4A, 4B and 4C, Table 4).

Five studies evaluated VE in directional sub-groups of NSLBP patients [9, 15, 16, 55, 56]. All five studies reported data for NSLBP related to postures and movements involving flexion, three also included extension-aggravated NSLBP patients [9, 16, 55], and one evaluated patients with passive extension-aggravated NSLBP [55].

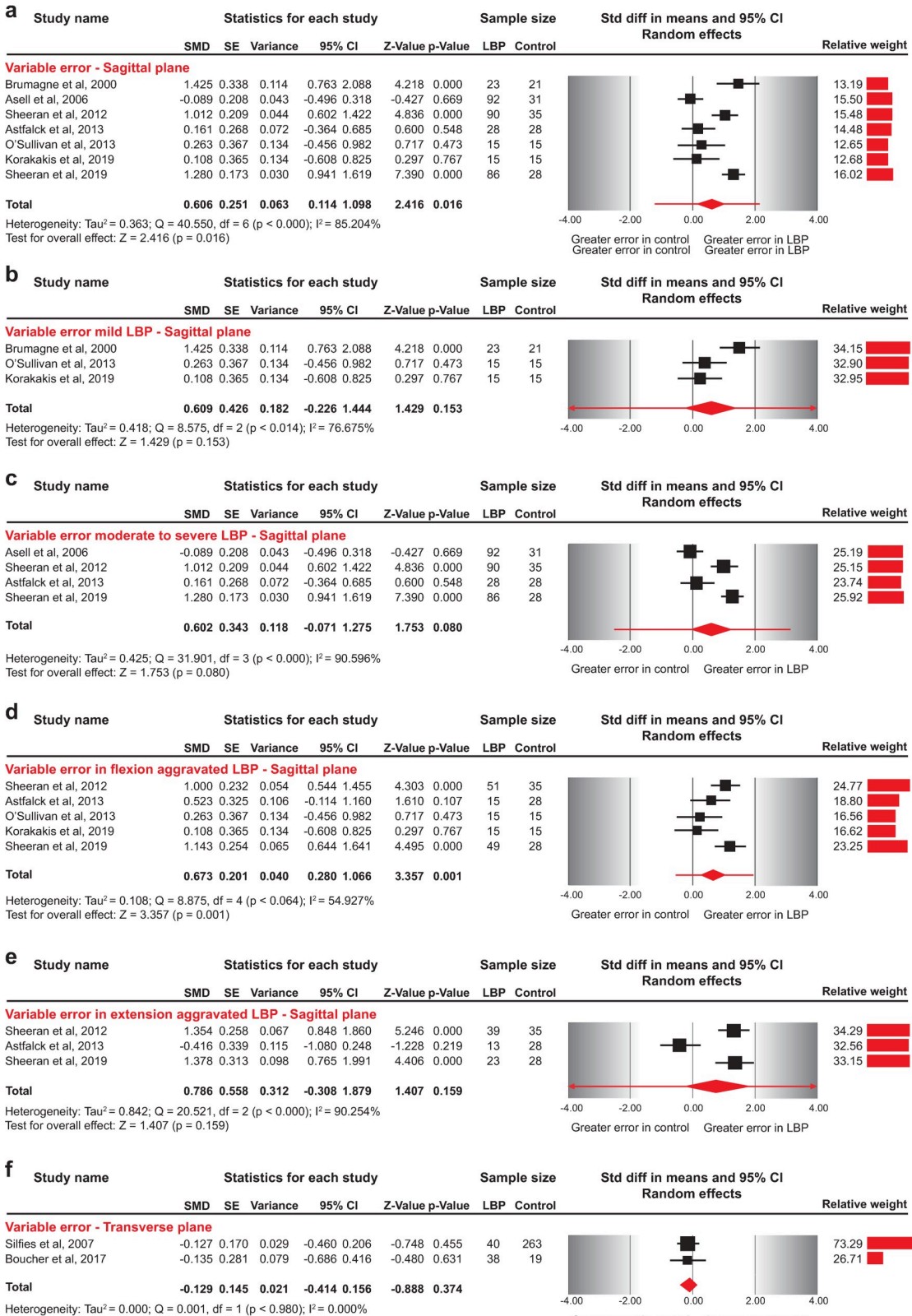

**Fig 4. Forest plots for variable error in sagittal and transverse plane repositioning between NSLBP and asymptomatic individuals.** a) Variable error between asymptomatic individuals and NSLBP patients, b) Variable error between asymptomatic

individuals and NSLBP patients with mild symptoms, c) Variable error between asymptomatic individuals and NSLBP patients with moderate to severe symptoms, d) Variable error between asymptomatic individuals and patients with flexion-aggravated NSLBP, e) Variable error between asymptomatic individuals and patients with extension-aggravated NSLBP, and f) Variable error between asymptomatic individuals and patients with NSLBP evaluated in transverse plane. The line crossing the summary effect estimate (diamond) in the forest plots represents the prediction interval, assuming that the effects are normally distributed [36]. Abbreviations: SMD, standardized mean difference; SE, standard error; CI, confidence intervals; NSLBP, non-specific low back pain.

Pooled results from five studies revealed a medium effect for larger VE for patients with flexion-aggravated NSLBP than asymptomatic individuals (SMD = 0.673, 95%CI: 0.280–1.066) for repositioning into a neutral sitting posture (Fig 4D). Removing studies that did not report VE in angular measures [15] or included adolescent patients [16] had no impact on the size of the effect estimate indicating a larger VE for patients with flexion-aggravated NSLBP (SMD = 0.756, 95%CI: 0.330–1.182; SMD = 0.694, 95%CI: 0.211–1.117, respectively) (Table 4).

Pooled results from three studies revealed no significant differences in VE between patients with extension-aggravated NSLBP and asymptomatic individuals (SMD = 0.786, 95%CI: -0.308 to 1.879) (Fig 4E). Excluding adolescent patients [16] had a significant impact on the direction and size of the effect estimate, indicating a greater (large effect) VE in AEP adult sub-group of NSLBP patients compared to asymptomatic individuals (SMD = 1.364, 95%CI: 0.974–1.754) (Table 4).

### Variable repositioning error–transverse plane

Two studies [20, 57] evaluated the VE of blindfolded NSLBP patients compared to asymptomatic individuals in the transverse plane and no significant differences were found in a task involving axial lumbar rotation (SMD = -0.129, 95%CI -0.414 to 0.156) (Fig 4F, Table 4).

**Constant repositioning error–sagittal plane.** Seven studies [9, 16, 55, 56, 61, 62] evaluated the error direction as a measure of bias of NSLBP patients compared to asymptomatic individuals in a sagittal plane repositioning task.

**Constant error between asymptomatic individuals and NSLBP patients.** Pooled data into a summary estimate revealed no difference in error direction between patients with NSLBP and asymptomatic individuals (SMD = -0.191, 95%CI -0.577 to 0.195) (Fig 5A). Removing one study [15] from data synthesis that did not report CE in angular measures, or one study [16] that included adolescents, did not impact on the direction of the effect estimate (SMD = -0.071, 95%CI: -0.433 to 0.291; SMD = -0266, 95%CI: -0.712,0.181, respectively) (Table 5).

### Constant error and subgroup analyses

By subgrouping studies according to NSLBP severity, we found a significant effect of symptoms' severity in error direction. Patients with mild NSLBP underestimated (medium effect) the target posture (SMD = -0.773, 95%CI: -1.271 to -0.276) compared to asymptomatic individuals. On the contrary, patients with moderate to severe NSLBP symptoms did not differ in CE compared to asymptomatic individuals (SMD = 0.140, 95%CI: -0.136 to 0.417) (Fig 5B and 5C, Table 5).

Five studies evaluated the error direction in directional sub-groups of NLBP patients [9, 15, 16, 55, 56]. All studies reported CE for NSLBP related to postures and movements involving flexion, three also included extension-aggravated NSLBP patients [9, 16, 55], and one evaluated patients with passive extension-aggravated NSLBP [55].

Pooled results from five studies revealed a small effect indicating that patients with flexion-aggravated NSLBP underestimated the target posture compared to asymptomatic

**Table 4. Summary of evidence for variable repositioning error.**

| Outcome | Comparisons | | Relative effect (95% CI) | Patients /controls (studies) | Quality of evidence (GRADE) | Evidence and significance |
|---|---|---|---|---|---|---|
| | Average estimate in NSLBP group | Average estimate in control group | | | | |
| **Variable error** Studies reporting VE in degrees | **NSLBP:** Pooled weighted mean ±SD not estimable* | **Control:** Pooled weighted mean±SD not estimable* | **SMD 0.655** [0.111, 1.200] Statistically significant difference | 334/158 (6) | ⊕◯◯◯ **Very low**[1,2,3] | Very low-level evidence of greater (medium effect) VE in NSLBP patients than in asymptomatic individuals |
| **Variable error** (adults only) | **NSLBP:** Pooled weighted mean ±SD not estimable* | **Control:** Pooled weighted mean±SD not estimable* | **SMD 0.681** [0.135, 1.227] Statistically significant difference | 321/145 (6) | ⊕◯◯◯ **Very low**[1,2,3] | Very low-level evidence of greater (medium effect) VE in adult NSLBP patients than in asymptomatic individuals |
| **Variable error** Mild NSLBP | **Mild-NSLBP:** Pooled weighted mean±SD not estimable* | **Control:** Pooled weighted mean±SD not estimable* | **SMD 0.609** [-0.226, 1.444] Non-statistically significant difference | 53/51 (3) | ⊕⊕◯◯ **Low**[2,3] | Low level evidence of no difference in VE between patients with mild NSLBP and asymptomatic individuals |
| **Variable error** Moderate to severe NSLBP | **Moderate-NSLBP:** Pooled weighted mean ±SD not estimable* | **Control:** Pooled weighted mean±SD not estimable* | **SMD 0.602** [-0.071, 1.275] Non-statistically significant difference | 296/122 (4) | ⊕◯◯◯ **Very low**[1,2,3] | Very low-level evidence of no difference in VE between patients with moderate to severe NSLBP and asymptomatic individuals |
| **Variable error** FP NSLBP subgroup | **FP-NSLBP:** Pooled weighted mean ±SD not estimable* | **Control:** Pooled weighted mean±SD not estimable* | **SMD 0.673** [0.280, 1.066] Statistically significant difference | 145/121 (5) | ⊕◯◯◯ **Very low**[1,2,3] | Very low-level evidence of greater (medium effect) VE in FP subgroup of NSLBP patients than in asymptomatic individuals |
| **Variable error** FP NSLBP subgroup (error in degrees) | **FP-NSLBP:** Pooled weighted mean ±SD was 4.1±2.7˚ (mean range 1.4 to 4.8) | **Control:** Pooled weighted mean±SD was 2.1 ±1.1˚ (mean range 1.3 to 2.8) | **SMD 0.756**[a] [0.330, 1.182] Statistically significant difference | 130/106 (4) | ⊕◯◯◯ **Very low**[1,2,3] | Very low-level evidence of greater (medium effect) VE in FP subgroup of NSLBP patients than in asymptomatic individuals |
| **Variable error** FP NSLBP subgroup (adults) | **FP-NSLBP:** Pooled weighted mean ±SD not estimable* | **Control:** Pooled weighted mean±SD not estimable* | **SMD 0.694** [0.211, 1.177] Statistically significant difference | 145/108 (4) | ⊕⊕◯◯ **Low**[2,3] | Low level evidence of greater (medium effect) VE in FP adult subgroup of NSLBP patients than in asymptomatic individuals |
| **Variable error** AEP NSLBP subgroup | **AEP-NSLBP:** Pooled weighted mean ±SD was 3.6±1.7˚ (mean range 2.2 to 3.9) | **Control:** Pooled weighted mean±SD was 2.2 ±1.2˚ (mean range 1.9 to 2.8) | **SMD 0.786** [-0.308, 1.879] Non-statistically significant difference | 75/91 (3) | ⊕◯◯◯ **Very low**[1,2,3] | Very low-level evidence of no difference in VE between AEP subgroup of NSLBP patients and asymptomatic individuals |
| **Variable error** AEP NSLBP subgroup (adults) | **AEP-NSLBP:** Pooled weighted mean ±SD was 3.2±1.5˚ (mean range 3.8 to 3.9) | **Control:** Pooled weighted mean±SD was 1.3 ±0.7˚ (mean range 1.9 to 1.9) | **SMD 1.364**[b] [0.974, 1.754] Statistically significant difference | 62/63 (2) | ⊕⊕⊕◯ **Moderate**[3] | Moderate level evidence of greater (large effect) VE in AEP adult subgroup of NSLBP patients than in asymptomatic individuals |
| **Variable error** Transverse plane | **NSLBP:** Pooled weighted mean ±SD was 1.7±0.7˚ (mean range 1.6 to 4.4) | **Control:** Pooled weighted mean±SD was 1.7 ±0.8˚ (mean range 1.6 to 3.6) | **SMD -0.129** [-0.414, 0.156] Non-statistically significant difference | 78/282 (2) | ⊕⊕◯◯ **Low**[1,3] | Low level evidence of no difference in transverse plane VE between NSLBP patients and asymptomatic individuals |

[1] Downgraded due to indirectness.

[2] Downgraded due to inconsistency.

[3] Downgraded due to imprecision.

* Pooled weighted mean not estimable as included studies did not report AE, VE, or CE as an angular measure, or did not report values for mean±SD.

[a] Pooled weighted SD was 2.0˚ resulting in an effect estimate of 1.5˚—non-clinically significant.

[b] Pooled weighted SD was 1.1˚ resulting in an effect estimate of 1.5˚—non-clinically significant.

Abbreviations: SMD, standardized mean difference, CI, confidence intervals; NSLBP, non-specific low back pain; SD, standard deviation; VE, variable error; NSLBP, non-specific low back pain; FP, flexion pattern; AEP, active extension pattern.

individuals (SMD = -0.408, 95%CI: -0.796 to -0.020) (Fig 5D). Removing one study [15] from data synthesis that did not report CE in angular measures had a significant impact on the effect estimate, resulting in no difference in CE between NSLBP patients and asymptomatic individuals (SMD = -0.298, 95%CI: -0.677 to 0.082). Finally, excluding adolescent patients with flexion-aggravated NSLBP [16] had no significant impact on the size of the effect estimate, indicating that patients with flexion-aggravated NSLBP underestimated the target posture compared to asymptomatic individuals (SMD = -0.562, 95%CI: -0.835 to -0.288) (Table 5).

The pooled results from three studies indicated no difference in CE between patients with extension-aggravated NSLBP compared to asymptomatic individuals (SMD = 0.538, 95%CI: -0.058 to 1.134) (Fig 5E). Excluding adolescent patients [16] did not affect the effect estimate (SMD = 0.746, 95%CI: -0.017 to 1.509) (Table 5).

One study [55] evaluating patients with passive extension-aggravated NSLBP presented very low certainty evidence of greater CE (target overestimation) in patients compared to asymptomatic individuals (MD = -8.600, 95%CI: -11.537 to -5.663) in repositioning into a neutral sitting posture.

One study [56] evaluating error direction in repositioning into a lordotic sitting posture presented very low certainty evidence of no difference in CE between patients with mild NSLBP and asymptomatic individuals with participants blindfolded (MD = -2.800, 95%CI: -5.622 to 0.022) or not (MD = -1.800, 95%CI: -4.495 to 0.895).

## Constant repositioning error–transverse plane

Very low certainty evidence from one study [57] showed that NSLBP patients overestimated the target position compared to asymptomatic individuals in a task involving axial lumbar rotation (MD = 0.850, 95%CI 0.643 to 1.057).

**Sensitivity analyses.** No study was judged as of "poor quality"; hence, no sensitivity analyses were conducted based on study quality.

**Absolute error.** Pooled results from eleven studies revealed a medium effect for greater AE in patients with NSLBP than asymptomatic individuals (SMD = 0.705, 95%CI: 0.199–1.212) (Fig 2A). By excluding two studies together [9, 55] or one at a time as outliers, the magnitude of the effect was decreased, but the AE remained greater in NSLBP patients than asymptomatic individuals (SMD = 0.405, 95%CI: 0.122–0.688; SMD = 0.601, 95%CI: 0.083–1.118; SMD = 0.559, 95%CI: 0.146–0.972, respectively).

Pooled results from seven studies according to NSLBP severity, revealed a significant effect of symptoms' severity in AE, with patients with moderate NSLBP presenting greater errors (medium effect) (SMD = 0.725, 95%CI: 0.007–1.443) (Fig 2B). By excluding two studies together [9, 55] or one at a time as outliers, had a significant effect on the effect estimate, presenting no difference in AE between NSLBP patients with moderate to severe symptom severity and asymptomatic individuals (SMD = 0.256, 95%CI: -0.125 to -0.636; SMD = 0.489, 95% CI: -0.150 to 1.128; SMD = 0.553, 95%CI: -0.264 to 1.370, respectively).

The pooled results from eight studies indicated a medium effect for greater AE for NSLBP patients than asymptomatic individuals (SMD = 0.876, 95%CI: 0.231–1.521) for the blindfolded repositioning task (Fig 3A). By excluding two studies together [9, 55] or only one study [55] as outliers, the magnitude of the effect was decreased, but NSLBP patients still presented greater AE (SMD = 0.448, 95%CI: 0.091–0.806; SMD = 0.663, 95%CI: 0.089–1.1237, respectively). Excluding only the study by Sheeran et al, [9] had a significant effect on the effect estimate, presenting no difference in AE between blindfolded NSLBP patients and asymptomatic individuals (SMD = 0.728, 95%CI: -0.025 to 1.481).

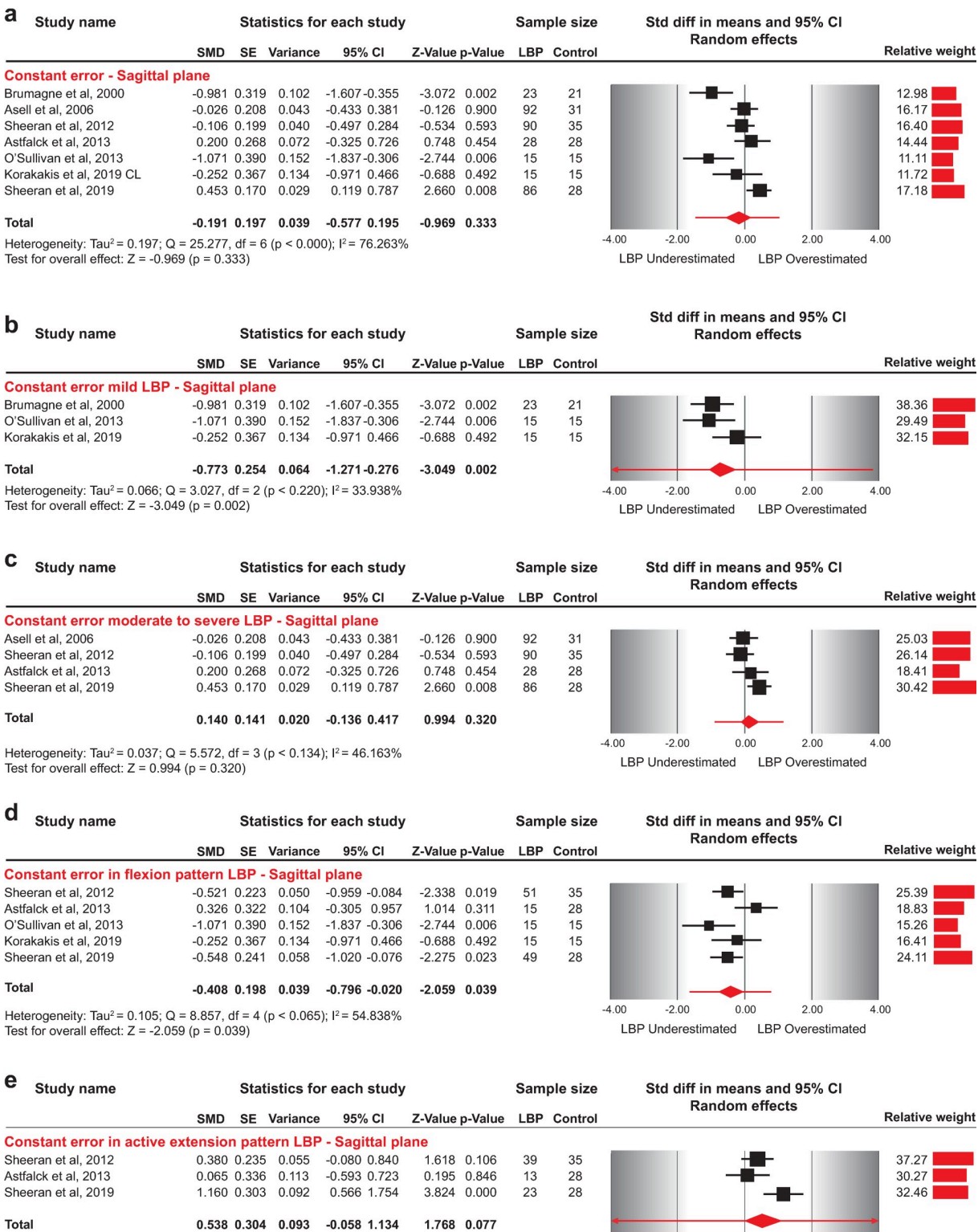

**Fig 5. Forest plots for constant error in sagittal plane repositioning between NSLBP and asymptomatic individuals.** a) Constant error between asymptomatic individuals and NSLBP patients, b) Constant error between asymptomatic individuals and NSLBP patients with mild symptoms, c) Constant error between asymptomatic individuals and NSLBP patients with moderate to severe symptoms, d) Constant error between asymptomatic individuals and patients with flexion-aggravated NSLBP, e) Constant error between asymptomatic individuals and

patients with extension-aggravated NSLBP. The line crossing the summary effect estimate (diamond) in the forest plots represents the prediction interval, assuming that the effects are normally distributed [36]. For data pooling, undershooting a target position was given a negative sign, whereas overshooting a target position was given a positive sign, resulting in changing the directionality of data in three studies [9, 55, 61]. Abbreviations: SMD, standardized mean difference; SE, standard error; CI, confidence intervals; NSLBP, non-specific low back pain.

**Table 5. Summary of evidence for constant repositioning error.**

| Outcome | Comparisons | | Relative effect (95% CI) | Patients /controls (studies) | Quality of evidence (GRADE) | Evidence and significance |
|---|---|---|---|---|---|---|
| | Average estimate in NSLBP group | Average estimate in control group | | | | |
| **Constant error** Studies reporting error in degrees | **NSLBP:** Pooled weighted mean ±SD not estimable* | **Control:** Pooled weighted mean±SD not estimable* | **SMD -0.071** [-0.433, 0.291] Statistically significant difference | 336/216 (6) | ⊕⊕◯◯ **Low[1,2]** | Low level evidence of no difference in error direction between patients with NSLBP and asymptomatic individuals |
| **Constant error** (Adults only) | **NSLBP:** Pooled weighted mean ±SD not estimable* | **Control:** Pooled weighted mean±SD not estimable* | **SMD -0.266** [-0.712, 0.181] Non-statistically significant difference | 321/145 (6) | ⊕⊕◯◯ **Low[1,2]** | Low level evidence of no difference in error direction between adult patients with NSLBP and asymptomatic individuals |
| **Constant error** Mild NSLBP | **Mild-NSLBP:** Pooled weighted mean±SD not estimable* | **Control:** Pooled weighted mean±SD not estimable* | **SMD -0.773** [-1.271, -0.276] Statistically significant difference | 53/51 (3) | ⊕⊕◯◯ **Low[2,3]** | Low level evidence that mild NSLBP patients underestimated (medium effect) the target posture compared to asymptomatic individuals |
| **Constant error** Moderate to severe NSLBP | **Moderate-NSLBP:** Pooled weighted mean ±SD not estimable* | **Control:** Pooled weighted mean±SD not estimable* | **SMD 0.140** [-0.136, 0.417] Non-statistically significant difference | 296/122 (4) | ⊕⊕◯◯ **Low[1,2]** | Low level evidence of no difference in error direction between patients with moderate to severe NSLBP and asymptomatic individuals |
| **Constant error** FP NSLBP subgroup | **FP-NSLBP:** Pooled weighted mean ±SD not estimable* | **Control:** Pooled weighted mean±SD not estimable* | **SMD -0.408** [-0.796, -0.02] Statistically significant difference | 145/121 (5) | ⊕◯◯◯ **Very low[1,2,3]** | Very low-level evidence that FP subgroup of NSLBP patients underestimated (small effect) the target posture compared to asymptomatic individuals |
| **Constant error** FP NSLBP subgroup (error in degrees) | **FP-NSLBP:** Pooled weighted mean ±SD not estimable* | **Control:** Pooled weighted mean±SD not estimable* | **SMD -0.298** [-0.677, 0.082] Non-statistically significant difference | 130/136 (4) | ⊕◯◯◯ **Very low[1,2,3]** | Very low-level evidence of no difference in error direction between FP subgroup of patients with NSLBP and asymptomatic individuals |
| **Constant error** FP NSLBP subgroup (adults) | **FP-NSLBP:** Pooled weighted mean ±SD not estimable* | **Control:** Pooled weighted mean±SD not estimable* | **SMD -0.562** [-0.835, -0.288] Statistically significant difference | 130/93 (4) | ⊕⊕◯◯ **Low[2,3]** | Low level evidence that adult FP subgroup of NSLBP patients underestimated (small effect) the target posture compared to asymptomatic individuals |
| **Constant error** AEP NSLBP subgroup | **AEP-NSLBP:** Pooled weighted mean±SD not estimable* | **Control:** Pooled weighted mean±SD not estimable* | **SMD 0.538** [-0.058, 1.134] Statistically significant difference | 75/91 (3) | ⊕◯◯◯ **Very low[1,2,3]** | Very low-level evidence of no difference in error direction between AEP subgroup of NSLBP patients and asymptomatic individuals |
| **Constant error** AEP NSLBP subgroup (adults) | **AEP-NSLBP:** Pooled weighted mean ±SD not estimable* | **Control:** Pooled weighted mean±SD not estimable* | **SMD 0.746** [-0.017, 1.509] Non-statistically significant difference | 62/63 (2) | ⊕⊕◯◯ **Low[2,3]** | Low level evidence of no difference in error direction between adult AEP subgroup of NSLBP patients and asymptomatic individuals |

[1] Downgraded due to indirectness

[2] Downgraded due to inconsistency

[3] Downgraded due to imprecision

* Pooled weighted mean not estimable as included studies did not report AE, VE, or CE as an angular measure, or did not report values for mean±SD

Abbreviations: SMD, standardized mean difference, CI, confidence intervals; NSLBP, non-specific low back pain; SD, standard deviation; CE, constant error; NSLBP, non-specific low back pain; FP, flexion pattern; AEP, active extension pattern.

**Variable error.**   The pooled data into a summary estimate from seven studies showed that patients with NSLBP have a greater variability (medium effect) in the RE about their mean response than asymptomatic individuals (SMD = 0.606, 95%CI 0.114–1.098) (Fig 4A). The exclusion of one study [62] as an outlier had a significant effect on the effect estimate, presenting no difference in VE between patients with NSLBP and asymptomatic individuals (SMD = 0.482, 95%CI: -0.042 to 1.006).

**Prediction intervals.**   The calculated prediction intervals describing the true effect size range included zero in all repositioning errors (Table 6).

## Discussion

### Main findings

These results demonstrated very low and low certainty evidence of greater RE in the sagittal plane and no difference in RE in the transverse plane between NSLBP patients and asymptomatic individuals with reference to a neutral sitting posture. Subgroup analyses suggested moderate certainty evidence of greater AE and repositioning variability between asymptomatic individuals and directional subgroups of NSLBP patients, but low and very low certainty evidence of variable results in error direction. Given the magnitude of error and the calculated "noise" of the measurement, we suggest that the statistically significant differences documented here, may be of limited clinical utility. Additionally, the calculated prediction intervals (true effect size range), included zero in all RE suggesting that these results should be interpreted with caution.

### Repositioning errors (pooling all NSLBP patients)

Low certainty evidence suggests no difference in error direction and significantly greater error variability (medium effect) between NSLBP patients and asymptomatic individuals. In contrast, AE demonstrated sensitivity to the measurement method, presenting either very low certainty of a medium effect for greater proprioceptive deficit among NSLBP patients or no difference with asymptomatic individuals. Measurement methods varied significantly among studies, with the majority using electromagnetic motion trackers [14, 16, 42, 43, 60, 61] and others using 3D motion analysis systems [9, 55, 56], tape measures [58], electronic goniometers [54, 62], custom-built apparatus [20, 57, 59], or strain gauge devices [15]. The shift in the direction of the effect by a single study [15] for example, suggests cautious inferences regarding AE magnitude and certainty in NSLBP patients, that could plausibly be attributed to the metric system used to express RE (percentage of strain gauge elongation relative to a referenced lumbar range of motion). When measuring small postural angular differences even the variability in retroreflective markers or electromagnetic sensors placement may significantly affect the measurement outcome. Interestingly, there was a lack of reliability data directly related to the setting, apparatus, population, spinal region, and task used in the studies. Only 5 studies [20, 56, 58, 61, 63] directly evaluated reliability matching the methodology used. The remaining of the studies only referred to previous reliability estimates: a) in a similar setting (i.e., in seated tasks using same apparatus) [9, 15, 55], b) with different tasks (i.e., standing, range of pelvic tilting, or perception of motion) [14, 16, 42, 43, 54, 57, 62], or c) no further information was available [59]. This methodological diversity it is likely that contributed to the heterogeneity observed in the quantitative synthesis of proprioceptive acuity indices.

Several other factors may have contributed to the observed heterogeneity, the inconsistency of findings, as well as the presence of outliers in the analyses in this systematic review. Namely, participant characteristics, lack of RE indices availability, and the specifics of the lumbar spine repositioning tasks. Only 6 studies reported an attempt to minimize selection bias by using

**Table 6. Prediction intervals calculated for pooled effect estimates.**

| ABOLUTE ERROR | | | |
|---|---|---|---|
| | | **Prediction interval** | |
| **Main comparison** | **SMD** | **Lower limit** | **Upper limit** |
| AE NSLBP-asymptomatic individuals | 0.705 | -1.1821 | 2.5921 |
| AE NSLBP-asymptomatic individuals (error in angular measure) | 0.651 | -1.6682 | 2.9702 |
| AE NSLBP-asymptomatic individuals (moderate to severe symptoms NSLBP) | 0.725 | -1.8122 | 3.2622 |
| AE NSLBP-asymptomatic individuals (mild symptoms LBP) | 0.525 | -1.4514 | 2.5014 |
| AE flexion-aggravated NSLBP-asymptomatic individuals | 1.219 | -0.4049 | 2.8429 |
| AE flexion-aggravated NSLBP-asymptomatic individuals (error in angular measure) | 1.287 | -1.8365 | 4.4105 |
| AE flexion-aggravated NSLBP-asymptomatic individuals (adults only) | 1.289 | -0.7075 | 3.2855 |
| AE extension-aggravated NSLBP-asymptomatic individuals | 1.486 | -14.1232 | 17.0952 |
| AE extension-aggravated NSLBP-asymptomatic individuals (adults only) | 2.080 | Non estimable | Non estimable |
| AE NSLBP-asymptomatic individuals (blindfolded participants) | 0.876 | -1.3786 | 3.1306 |
| AE NSLBP-asymptomatic individuals (vision available) | 0.325 | -0.8812 | 1.5312 |
| AE NSLBP-asymptomatic individuals (Task in transverse plane) | 0.665 | -13.968 | 15.298 |
| VARIABLE ERROR | | | |
| | | **Prediction interval** | |
| **Comparison** | **SMD** | **Lower limit** | **Upper limit** |
| VE NSLBPasymptomatic individuals | 0.606 | -1.0718 | 2.2838 |
| VE NSLBP-asymptomatic individuals (error in angular measure) | 0.655 | -1.2491 | 2.5591 |
| VE NSLBP-asymptomatic individuals (adults only) | 0.681 | -1.2115 | 2.5735 |
| VE NSLBP-asymptomatic individuals (mild symptoms NSLBP) | 0.609 | -9.2290 | 10.447 |
| VE NSLBP-asymptomatic individuals (moderate to severe symptoms NSLBP) | 0.602 | -2.5683 | 3.7723 |
| VE flexion-aggravated NSLBP-asymptomatic individuals | 0.673 | -0.5522 | 1.8982 |
| VE flexion-aggravated NSLBP-asymptomatic individuals (error in angular measure) | 0.756 | -0.9228 | 2.4348 |
| VE flexion-aggravated NSLBP-asymptomatic individuals (adults only) | 0.694 | -1.2905 | 2.6785 |
| VE extension-aggravated NSLBP-asymptomatic individuals | 0.786 | -12.8575 | 14.4295 |
| VE extension-aggravated NSLBP-asymptomatic individuals (adults only) | 1.364 | Non estimable | Non estimable |
| VE NSLBP-asymptomatic individuals (Task in transverse plane) | -0.129 | Non estimable | Non estimable |
| CONSTANT ERROR | | | |
| | | **Prediction interval** | |
| **Comparison** | **SMD** | **Lower limit** | **Upper limit** |
| CE NSLBP-asymptomatic individuals | -0.191 | -1.4392 | 1.0572 |
| CE NSLBP-asymptomatic individuals (error in angular measure) | -0.071 | -1.2328 | 1.0908 |
| CE NSLBP-asymptomatic individuals (adults only) | -0.266 | -1.7534 | 1.2214 |
| CE NSLBP-asymptomatic individuals (mild symptoms LBP) | -0.773 | -5.3595 | 3.8135 |
| CE NSLBP-asymptomatic individuals (moderate to severe symptoms LBP) | 0.140 | -0.8870 | 1.1670 |
| CE flexion-aggravated NSLBP-asymptomatic individuals | -0.408 | -1.6164 | 0.8004 |
| CE flexion-aggravated NSLBP-asymptomatic individuals (error in angular measure) | -0.298 | -1.7158 | 1.1198 |
| CE flexion-aggravated NSLBP-asymptomatic individuals (adults only) | -0.562 | -1.1632 | 0.0395 |
| CE extension-aggravated NSLBP-asymptomatic individuals | 0.538 | -6.2508 | 7.3268 |
| VE extension-aggravated NSLBP-asymptomatic individuals (adults only) | 0.746 | Non estimable | Non estimable |

Abbreviations: AE, absolute error; VE, variable error; CE, constant error; SMD, standardized mean difference; NSLBP, non-specific low back pain.

matching criteria [14–16, 43, 56, 61], while half of the studies recruited relatively small sample sizes [14, 15, 43, 54, 56, 58, 59, 62]. A considerable variation could be observed in the chronicity of NSLBP of the participants, ranging from 4 weeks to more than 3 months, while in 4

studies [20, 58, 60, 62] this inclusion criterion was not clearly defined. Similarly, contradictory inclusion criteria were used for the asymptomatic individuals, with studies recruiting participants that had not experienced NSLPB at all, not in the last one or two years, or simply if they were asymptomatic the day of testing. The diversity in demographic characteristics along with the missingness of relevant information could plausibly explain the presence of outliers in quantitative synthesis and the inconsistency of the findings. To illustrate, the study of Brumagne et al, [62] was an outlier in VE and CE analyses; however, they recruited significantly younger individuals (mean age 22 years) than all other included studies (participant age >30 years) and provided no information regarding the chronicity of NSLBP and the NSLBP status–current or previous—for the asymptomatic individuals. From a different perspective, the RE indices reported limited a comprehensive evaluation of proprioceptive acuity in NSLBP. Most studies reported mainly one aspect of RE (15/16 studied AE), which constrained the analysis possible for error direction or the inconsistency about the target posture [25]. Finally, inconsistencies in measurement approaches likely affected the quantitative synthesis and complicated interpretation of the findings. Trial repetitions ranged from 3 to 10, with half of studies using 5 or more attempts [14, 20, 54, 56, 57, 60–62]. While in 4 studies [15, 16, 43, 56] participants were allowed to warm-up, were given practice trials, and the repositioning task was demonstrated, for the remaining only warm-up or practice trials were performed. On the one hand, the stability of reliability indices has been argued to be dependent on the number of attempts used to calculate them and precision estimates are underestimated for data derived from three or fewer attempts, such that at least five attempts are suggested [64]. On the other hand, evidence suggests that practice improves performance in a positioning task [65] and when given a sufficient number of learning trials, NSLBP patients were able to reproduce a posture with the precision and variability observed in asymptomatic individuals [66]. These issues make it difficult to ascertain precisely the factors that contributed to the variability observed. By subgrouping according to symptoms' severity, low and very low certainty evidence suggests no difference in error variability, significantly greater AE in NSLBP patients, and inconsistent results regarding error direction. The lack of a dose-response relationship between proprioceptive deficit and pain and disability [19] can probably explain this variation and the significant or not effects can be attribute to sample characteristics, or diversity in methodology implemented in included studies.

Very low certainty evidence suggests greater AE (medium effect) between NSLBP patients and asymptomatic individuals in a blindfolded repositioning task, and no difference with vision available. Despite this finding seeming logical, it is conflicting with previous work [67–69]. Study methodological diversity render firm conclusions unsafe.

## Repositioning errors in directional subgroups of NSLBP patients

Low and very low certainty evidence suggests greater AE (large effect) and VE (medium effect), and a tendency (small effect) to underestimate the neutral target posture in patients (combined adults and adolescents) with flexion-aggravated NSLBP compared to asymptomatic individuals. The visual observation of the forest plots strengthens the notion that NSLBP is not a homogenous group. Three studies [9, 14, 55] that specifically used the O'Sullivan's classification system (i.e., flexion-aggravated NSLBP subgroup) appeared as outliers when analyzed along with all other studies or in the moderate NSLBP subgroup analysis (AE and VE), suggesting that this subgroup may differ across several variables with other NSLBP subgroups.

The results for AE in patients with flexion-aggravated NSLBP demonstrated a relatively homogeneous pattern for less accuracy. The exclusion from quantitative synthesis of adolescents did not affect the evidence that the REs are greater in NSLBP as compared to

asymptomatic individuals. The fact that the flexion-aggravated NSLBP subgroup is character-ized by adult and adolescent patients reporting sitting as a pain provoking activity, may plausi-bly explain the large effect of the proprioceptive deficit. However, the VE seemed to associate with the severity of symptoms and disability among studies, with increased nociceptive "noise" increasing variability of repositioning [9, 16, 55]. These findings indicate that the reduced accuracy and increased variability are common characteristics of patients with flexion-aggra-vated NSLBP, irrespective of age. Interestingly, the flexion-aggravated adult NSLBP subgroup demonstrated a direction-specific neutral spine position deficit (CE) and a tendency to reposi-tion in a pain-provoking direction (undershooting). A pattern that was not evident in adoles-cents with flexion-aggravated NSLBP, who tended to overshoot the target posture. It has been argued that the sensation of pain should reinforce patients' desire to adopt postures away from the pain provoking movement [9], an assumption that was not confirmed in adult patients with flexion-aggravated NSLBP. It has been hypothesized that the underestimation of target posture and the observed abdominal muscle hyperactivity might be maladaptive by not allow-ing them to deviate from the flexed posture [9]. Evidence from adolescents with flexion-aggra-vated NSLBP suggests that levels of muscle activation did not differ as compared with asymptomatic individuals [10], indicating a significant difference with adult patients. None-theless, the generalizability of this finding is limited due to the contamination of the adolescent sample by the combination of adolescents with "multidirectional" and flexion-aggravated NSLBP [16].

In contrast, moderate certainty evidence suggests greater AE (very large effect) and VE (large effect) in adults with extension-aggravated NSLBP compared to asymptomatic adults. Low-level certainty evidence presented no difference in error direction irrespectively of age in patients with extension-aggravated NSLBP compared to asymptomatic individuals. The inclusion or not of adolescents in quantitative synthesis had a significant impact on the direction of the effect esti-mate by increasing the magnitude of both AE and VE in the adult subgroup. Adolescents with extension-aggravated NSLBP presented better accuracy and less variability compared to adults. This finding contrasts with evidence indicating that the proprioceptive performance of adoles-cents is less efficient than that of adults [70, 71] and must be elucidated in future research. Addi-tionally, adolescents with extension-aggravated NSLBP presented better accuracy and less variability compared to the flexion-aggravated adolescent NSLBP subgroup. It has been argued that the greater report of pain in the lumbar spine and the sitting as the most provoking posture in those adolescents in the flexion sub-group could explain this discrepancy [16].

Similar to flexion-aggravated NSLBP, the extension-aggravated adult NSLBP patients repo-sitioned into a pain provoking direction (overshooting). The adolescents in the extension pat-tern did not differ to asymptomatic individuals in error direction. Given the sparse evidence in spinal proprioceptive acuity in adolescents with NSLBP, these results should be interpreted with caution.

## Clinical implications and methodological considerations

Mounting evidence suggests that NSLBP is not a homogenous group, but rather represents a variety of clinical presentations which may differ across numerous domains such as physical, psychological and lifestyle aspects [4–6, 72]. Postural training approaches involving spinal repositioning sense [73], or targeting postural and movement behaviors [74] have been advo-cated to reduce pain and disability. While the findings of our review support the contention that proprioceptive acuity is impaired among people with NSLBP, we must ask how meaning-ful the observed difference is considering the aforementioned concerns regarding how confi-dent we can be about the findings.

In practice, the MCID for any measure will be the sum of the "noise" in the measurement (smallest real difference) and that amount deemed clinically important, for the particular situation. The amount deemed clinically important will be influenced by the specifics of the situation at hand. For example, in this review the SMDs for RE between asymptomatic and NSLBP patients ranged from 1.5˚ to 4.4˚, which has clinical importance in terms of discriminating patients from asymptomatic individuals. The measurement error reported in included studies was >5˚, hence the MCID would be somewhere between 6.5˚ and 9.4˚. Therefore, we suggest that despite the statistically significant differences in AE, VE, and CE and the potential to detect such small changes, the clinical applicability of measuring such small impairments is debatable. Previous research [73] evaluating a guided postural intervention which reported changes in RE all being <5˚ support this contention.

## Limitations and future research

In the light of contemporary evidence highlighting that discrepancies exist between risk of bias and study quality assessment findings, with the former impairing accurate inferences about the credibility of study outcomes [75, 76] we deviated from our published analysis plan (PROSPERO). Also, due to variability of data reporting in included studies and the impact on pooling in previous systematic reviews [11, 12], we changed the meta-analysis software used for quantitative synthesis.

We acknowledge the limitations of pooling results from different measurement methods; however, subgroup analyses gave insight into this discrepancy. Another limitation is the arbitrarily selected cut-off values to subclassify participants with mild or moderate to severe NSLBP.

Despite the notion that 3D software-based devices are more accurate than other measures, the increased variability in setting and the involvement of testers may have affected their precision. We recommend future studies to report within- and between-day reliability, and the measurement errors of both devices and approaches in order to make meaningful inferences of repositioning acuity. Methodological diversity, heterogeneity, large and inconclusive prediction intervals, and unjustified "noise" in the measurement hamper generalisability and render firm conclusions unsafe.

Further research is needed to evaluate the acceptable degree of error in lumbo-pelvic proprioceptive testing considering factors such as a) the absolute value of the measurement, b) the available motion of the segment, c) the degree of difference observed between people with or without pain, or d) the degree of change observed from the start to the end of rehabilitation.

## Conclusions

The current review demonstrated very low and low certainty evidence of greater seated sagittal plane RE in NSLBP patients compared to asymptomatic individuals. Subgroup analyses suggested moderate certainty evidence of greater AE and repositioning variability between asymptomatic individuals and directional subgroups of NSLBP patients, but low and very low certainty evidence of variable results in error direction. Given the magnitude of error and the calculated "noise" of the measurement, we suggest that the statistically significant differences documented here, may be of limited clinical utility.

## Supporting information

**S1 Checklist.**
(DOC)

**S1 File. Underlying data set.**
(XLSX)

**S2 File. Electronic database search strategy in PubMed.**
(PDF)

## Author Contributions

**Conceptualization:** Vasileios Korakakis.

**Data curation:** Vasileios Korakakis, Kieran O'Sullivan, Argyro Kotsifaki, Yiannis Sotiralis, Giannis Giakas.

**Formal analysis:** Vasileios Korakakis, Kieran O'Sullivan, Argyro Kotsifaki.

**Funding acquisition:** Vasileios Korakakis.

**Investigation:** Vasileios Korakakis, Argyro Kotsifaki.

**Methodology:** Vasileios Korakakis, Kieran O'Sullivan, Argyro Kotsifaki, Yiannis Sotiralis, Giannis Giakas.

**Project administration:** Vasileios Korakakis.

**Software:** Vasileios Korakakis.

**Supervision:** Giannis Giakas.

**Validation:** Vasileios Korakakis.

**Visualization:** Vasileios Korakakis.

**Writing – original draft:** Vasileios Korakakis, Kieran O'Sullivan, Argyro Kotsifaki, Yiannis Sotiralis, Giannis Giakas.

**Writing – review & editing:** Vasileios Korakakis, Kieran O'Sullivan, Argyro Kotsifaki, Yiannis Sotiralis, Giannis Giakas.

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
