## [Decision Letter · Decision Letter 0]

26 Jan 2021

PONE-D-20-25447

Lumbo-pelvic proprioception in sitting is impaired in subgroups of low back pain – but the clinical utility of the differences is unclear. A systematic review and meta-analysis

PLOS ONE

Dear Dr. Korakakis,

Thank you for submitting your manuscript to PLOS ONE. After careful consideration, we feel that it has merit but does not fully meet PLOS ONE’s publication criteria as it currently stands. Therefore, we invite you to submit a revised version of the manuscript that addresses the points raised during the review process.

We look forward to receiving your revised manuscript.

Kind regards,

José M. Muyor

Academic Editor

PLOS ONE

2.) Thank you for including the statement that "PubMed, Cochrane, CINAHL, EMBASE and Web of Science databases were independently searched by two reviewers (VK and YS) from inception of database to 28 March 2020 without language restriction, with aim to reduce language and publication bias. ". Please revise this statement to clarify whether all databases were searched from inception, or if there were any limits placed on the publication dates in your search.

3.) In the methods, please describe how risk of bias was assessed in individual studies (including specification of whether this was done at the study or outcome level, or both, and the specific test employed, such as the I^2 statistic), and how this information was used in any data synthesis.

4.) In your Data Availability statement, you have not specified where the minimal data set underlying the results described in your manuscript can be found. PLOS defines a study's minimal data set as the underlying data used to reach the conclusions drawn in the manuscript and any additional data required to replicate the reported study findings in their entirety. All PLOS journals require that the minimal data set be made fully available. For more information about our data policy, please see http://journals.plos.org/plosone/s/data-availability.

5.) Please note that in order to use the direct billing option the corresponding author must be affiliated with the chosen institute. Please either amend your manuscript to change the affiliation or corresponding author, or email us at plosone@plos.org with a request to remove this option.

6.) Please include captions for your Supporting Information files at the end of your manuscript, and update any in-text citations to match accordingly. Please see our Supporting Information guidelines for more information: http://journals.plos.org/plosone/s/supporting-information.

Reviewers' comments:

Reviewer's Responses to Questions

**Comments to the Author**

1. Is the manuscript technically sound, and do the data support the conclusions?

Reviewer #1: Yes

Reviewer #2: Yes

2. Has the statistical analysis been performed appropriately and rigorously? 

Reviewer #1: Yes

Reviewer #2: Yes

3. Have the authors made all data underlying the findings in their manuscript fully available?

Reviewer #1: Yes

Reviewer #2: Yes

4. Is the manuscript presented in an intelligible fashion and written in standard English?

Reviewer #1: Yes

Reviewer #2: Yes

5. Review Comments to the Author

Reviewer #1: The authors have prepared a thoughtful and necessary review on lumbopelvic hip proprioception in a sitting posture in non-specific low back pain. This is a great update to review and provide a meta-analysis of the seated outcomes since the most recent Tong et al, 2017 review that was mixed on standing and seated proprioception. The inclusion of subgrouping as a secondary aim allowed for some unique analyses that shed light on clinical application.

Abstract: Well organized and informative, although clarity on use of “low back pain” as the same as “non-specific low back pain” is important to define from the outset. This would also strengthen the inclusion of the secondary objective of the review.

Sitting is not mentioned in the objective or as part of the active repositioning in the search strategy portion of the methods. As this is central to the review and meta-analyses (and in the title) the findings and quality of evidence should both be included here along with this.

Directional subgrouping pattern needs a definition (may not have space within abstract, but is assumed throughout review)

Lines 55-62: The introduction begins with the same sentences as the abstract, except (NSLBP) is included in the introduction. These sentences should not be an exact copy. [The inclusion of this abbreviation addresses the confusion of using NSLBP and LBP interchangeably earlier in the abstract.] A brief definition of NSLBP would be helpful to include in the opening paragraph, perhaps with the sentence indicating "proprioceptive deficits might be associated with the underlying characteristics and mechanisms of NSLBP development”. The connection between the two and need for the review could be stressed further beyond the described lack of separation of seated proprioceptive assessment and intervention in prior reviews. This will strengthen and could link to the clinical implication discussion at the end of the review surrounding the MCID.

Lines 69-72: Including the level of evidence of these studies would further strengthen this statement.

Lines 96-97: Item (a) reads as exclusion criteria, could be reworded simply as “systematic reviews and case studies were excluded”. Were case series also excluded?

Lind 104: Listing some of the serious pathologies from the Tong review referenced would be useful for clarity especially with a diagnosis of exclusion (NSLBP) as the focus of the review.

Lines 151-152: Can the authors add a reference for the SMD effect scale?

Lines 175-183: I agree with the authors reasoning behind the MCID establishment, although including other references that assisted in arriving at this conclusion would bolster the decision-making process that is fairly important for this review. This links back to my earlier comment from the introduction and to a later comment from the discussion. Establishing the reasoning that is given (more context was given in the discussion for this decision) throughout gives stronger purpose to this review overall.

Lines 189-197: The current sentence structure is difficult to follow. If it was broken into two sentences explaining criteria for downgrading in the second sentence, it may be easier to follow.

Lines 202-203: Figure 1 explains the process well, but this sentence is somewhat misleading by indicating the search went from 3,137 and after duplicates were removed, only 16 remained that met inclusion. This could be reworded to better explain what Figure 1 states about the authors did throughout the review process to ultimately get to the 16 included studies, or simply direct the reader to Figure 1.

Lines 216-217: The BMI or body fat % difference is mentioned from 2 of the studies, but not included in Table 1 that I can see in the total sample column. The values associated with the differences, although not primary outcomes, since they are highlighted here should be included. The age difference mentioned across 2 studies could easily be noted in Table 1 as means are included already. This becomes important with adolescent analysis later on. May want to indicate adolescent values specifically due to that separation in analyses?

Line 223: Recent history for controls varied across the included studies just as the NSLBP timing and at least a range of time would be beneficial to match the robust level that the NSLBP inclusion was explored in the following sentence (which was well done).

Lines 241-244: How were the Lam and Maffey-Ward articles considered in this part of the assessment since they were not designed for a group comparison? These seem to be a clear exception to the rule in many aspects, which is addressed well in some ways (lines 271-219, Table 2) but not clear in others since they are different studies by the same group merged for analysis.

Lines 252-260: Separating out with "Included and excluded studies” subheadings for each outcome becomes very tedious throughout the remainder of the review. The sentence(s) that state how many studies were included in each section could simply be stated at the beginning of each section without that additional subheading to streamline the sections. It is not consistent throughout each subheading, so removing it altogether from the sections where it is present could be helpful.

Lines 263-264: Mild or moderate NSLBP symptoms are used to subgroup and mild was defined with Oswestry Disability Index and Roland Morris scores,( <15 ODI and <5 Roland Morris) however moderate was not clearly defined with a score range. Also, would a mild rating include 0 on either questionnaire since that is less than 15 and 5?

Line 282: The authors begin excluding adolescents and it is unclear if this is another aim altogether at this point or done intentionally for organization with the subheadings. Perhaps if the age differences were highlighted in the tables as they are mentioned earlier, this exclusion from analysis to show significant effects would be clear without an extensive additional explanation.

The discussion is touched on with the discrepancies between adolescents and adults, (lines 550-553, 559-563) yet could be emphasized based on findings within directional subgroup.

Lines 378-379: Throughout the results, as there are many outcomes to describe, the structure becomes very redundant. The authors do a very nice job in many places of describing directional results in active voice, although some places only state significance or an effect, which is only part of the story. The impactful part of the results is the directional piece and understanding the overall findings for each outcome after the analyses. Removal of the unnecessary subheadings will help with the choppiness of these sections as well.

Line 583: This is a major highlight and unclear why MCID calculation could happen in discussion based on some of the data, but was not formally included earlier. Although not all studies included had reliability data, there were other outcomes reported that were incomplete.

Discussion: An extended discussion based on methodological discrepancies between NSLBP and controls primary aim and the reasons that led to outliers in each analysis would enrich the section. I agree with the authors when they state methodological diversity made concrete conclusions unsafe, although further discussion of why and how that happened with the compilation of these 16 studies could happen. Although the evidence revealed very low and low certainty in the primary aim, the discussion of this matter pulling examples from more of the included studies would improve the discussion section. The participants, interventions, measurement, specific outcomes, reporting differences and even lack of data availability played into the decisions in each analysis and ultimately affected the SMDs. The focus seemed shift heavily to the subgroup secondary aim, which was great information but not the primary aim of the review.

Reviewer #2: Overall, this is an interesting article. Some minor issues need to be resolved:

INTRODUCTION:

- The introduction is concrete, which is appreciated. However, it would be advisable to include a paragraph about the systematic reviews that have been done on low back pain, and the novelty of the present review compared to previous ones.

- On the other hand, it is advisable that when it is mentioned that a systematic review has been done, "with meta-analysis" should be included (e.g., line 73).

METHOD:

- The method is very specific, which is appreciated as it allows for replicability of the study. However, it is advised to include in the manuscript the keywords and search formula in the databases, including Boolean operators.

RESULTS:

- The results section includes a lot of information, which is normal in a systematic review with meta-analysis, but needs to improve in order and summarise the information presented in order to be easily followed. Thus, it is advisable to make a first table (Table 1) where by thematic areas the reader can see what has been done in each study, excluding the results of this table, as in fact they are not the expected results of a meta-analysis and the results analysed by the meta-analysis are perfectly shown in the following tables. It is recommended for this table 1 to be divided into section by subject.

DISCUSSION:

- The discussion is adequate to the results and the bibliographic depth required for explanations to the findings.

CONCLUSION:

- The conclusion is adequate and responds to the objective.

---

## [Decision Letter · Decision Letter 1]

12 Apr 2021

Lumbo-pelvic proprioception in sitting is impaired in subgroups of low back pain – but the clinical utility of the differences is unclear. A systematic review and meta-analysis

PONE-D-20-25447R1

Dear Dr Korakakis,

We’re pleased to inform you that your manuscript has been judged scientifically suitable for publication and will be formally accepted for publication once it meets all outstanding technical requirements.

Kind regards,

José M. Muyor

Academic Editor

PLOS ONE

Reviewer's Responses to Questions

**Comments to the Author**

1. If the authors have adequately addressed your comments raised in a previous round of review and you feel that this manuscript is now acceptable for publication, you may indicate that here to bypass the “Comments to the Author” section, enter your conflict of interest statement in the “Confidential to Editor” section, and submit your "Accept" recommendation.

Reviewer #1: All comments have been addressed

Reviewer #2: All comments have been addressed

2. Is the manuscript technically sound, and do the data support the conclusions?

Reviewer #1: Yes

Reviewer #2: Yes

3. Has the statistical analysis been performed appropriately and rigorously? 

Reviewer #1: Yes

Reviewer #2: Yes

4. Have the authors made all data underlying the findings in their manuscript fully available?

Reviewer #1: Yes

Reviewer #2: Yes

5. Is the manuscript presented in an intelligible fashion and written in standard English?

Reviewer #1: Yes

Reviewer #2: Yes

6. Review Comments to the Author

Reviewer #1: Thank you to the authors for the great effort placed toward the revisions. They were all well addressed and the manuscript is very impactful and clear!

Reviewer #2: Thank you for the changes made. All issues have been addressed by the authors. In my opinion, the article can be published in its current state.

---

## [Editor Report · Acceptance letter]

14 Apr 2021

PONE-D-20-25447R1 

Lumbo-pelvic proprioception in sitting is impaired in subgroups of low back pain – but the clinical utility of the differences is unclear. A systematic review and meta-analysis 

Dear Dr. Korakakis:

I'm pleased to inform you that your manuscript has been deemed suitable for publication in PLOS ONE. Congratulations! Your manuscript is now with our production department. 

Kind regards, 

on behalf of

Dr. José M. Muyor 

Academic Editor

PLOS ONE